# Considering distributive justice as a planning principle helps navigate a diversity of future energy infrastructure designs

Katherine Emma Lonergan [1,2] & Giovanni Sansavini [1,2] ✉

The low-carbon energy transition could receive broader societal support if it also delivers a more just distribution of burdens and benefits. However, achieving more just systems in practice is complicated by contested interpretations of justice, conflicting system impacts, and overarching technical system constraints. To resolve these challenges, we develop a set of indicators for distributive justice based on justice theory and current policy to assess low-carbon and low-cost European energy system designs. We find that accounting for distributive justice can help narrow the field of technically viable system design choices without introducing substantive cost trade-offs. Considering public opinion to create a balanced approach to distributive justice produces more consistent continental-level technology capacity recommendations than provided by theory alone, showing the value of considering public opinion in systems-level energy planning. Our results support policymakers in enabling a sustainable energy transition that is also just.

The low-carbon energy transition offers multiple opportunities to create a more just society. Globally, decarbonising the energy system is key to mitigating the worst effects of climate change and protecting the world's most vulnerable from climate impacts they did not cause[1]. Locally, a shift towards renewable energy can provide new employment, foster regional growth, and improve air quality[2]. The decentralised nature of renewables invites a wide set of stakeholders to participate in energy decision-making; for instance, through the installation of at-home solar photovoltaics (PV). Renewables' wide deployment potential also provide broad choice in how to build future energy systems[3] and independence from entrenched patterns of fossil fuel reliance[4].

Capitalising on the opportunity to create more just societies through energy systems development can occur via several routes. For example, creating inclusive and accessible decision-making processes, such as via public consultation processes, supports procedural justice[5] and mitigates the risks of individual rights being trampled within the development process. Identifying who is affected by specific decisions supports recognition justice and can proactively help avoid creating new injustices[6]. Building supply chains where human rights are respected throughout supports cosmopolitan justice, while using energy systems development to correct prior injustices supports restorative justice[7].

One unresolved issue is how to fairly allocate the many burdens and benefits associated with energy systems. On one side, new energy infrastructure provides investment opportunity, employment, and, depending on the technology and operational scheme, energy autonomy[8]. On the other side, installing new technologies can also be costly, require greater land use, and result in job losses for the fossil fuel sector[8]. The quest for distributive justice is highly policy-relevant given that many energy system impacts are tangible to everyday people, like energy prices, land use, and employment opportunities. Other transition impacts are more personal but no less policy-relevant: for instance, changes to community sense of place and to personal identities linked to the growth[9] and decline[10] to specific industries can lead to tense political stand-offs. Philosophers have long argued about what it means achieve distributive justice[11,12], but answering "What is just?" is a highly subjective question whose answer might change over time, even for an individual.

In the context of infrastructure planning, existing legal frameworks and consultation processes provide a basis upon which to make

[1]Reliability and Risk Engineering, ETH Zurich, Zurich, Switzerland. [2]Institute of Energy and Process Engineering, ETH Zurich, Zurich, Switzerland.
✉e-mail: sansavig@ethz.ch

just decisions[13]. However, these processes are time-consuming[13] and ill-suited to navigating the planning complexities[14] associated with many possible future energy systems, especially in terms of understanding the quantifiable distributional impacts of future energy systems[15]. Energy system models offer one route forward. Energy scientists[16] have recently joined climate scientists[17,18] in exploring the potential of justice indicators or metrics for identifying just, or comparatively more just, decisions in a modelling framework. The crux of the argument for using justice indicators in both communities is that indicators provide a comparative basis for decision-making. By extension, models can help identify the design choices that best fulfil a given objective, even where an ideal solution cannot be identified. For example, a no-cost energy system is impossible, but models can identify the least-cost solution. Incorporating models into decision-making can, therefore, facilitate practical decision-making[11]. Considering justice within an energy systems model is advantageous since models can generate a wide variety of energy system designs and manage impact trade-offs, even in complex, uncertain decision-making scenarios[14]. These candidate designs can then be systematically assessed for distributional impacts[19–21].

Modellers are exploring potential social impacts of future system design and working to overcome historical critiques that models lack consideration of social outcomes[14,22]. For example, Sasse and Trutnevyte[21] apply a subnational electricity system model to understand how a low-carbon transition could affect regional European inequalities, considering particulate matter emissions, land use, electricity prices, and employment opportunities. Other recent studies focus on more specific issues like future employment opportunities, such as investigations for European fossil fuel workers[23], European low-carbon hydrogen production[24], and American power sector employment[25], the latter of which also accounted for gender-based impacts.

These efforts support the need to proactively consider justice in energy infrastructure planning, given the scale and urgency of the low-carbon transition, as well as new policy initiatives targeting just

development[8,26]. The European Union (EU), for example, has policies in place to be the first net-zero continent and to support a fair transition[27–29]. The bloc is aiming to fast-track regulatory processes for energy infrastructure approval to combat increased commissioning times[30,31] and waning emissions budgets[32]. However, moving forward more quickly must not come at the expense of just outcomes, given that the EU is committed to "not leaving anybody behind"[28] and that social acceptance influences the speed of infrastructure development[33,34].

Providing helpful policy input requires justice-informed modelling efforts to link concepts of distributive justice to technical system designs. To maximise their utility, justice-informed planning methods should also be able to: (1) quantify the trade-offs of pursuing one vision of justice versus another, (2) provide robust recommendations for capacity requirements, and (3) account for prevailing interpretations of what it means to be fair.

To help close this gap, we develop a set of justice indicators and test their ability to refine the design space suggested by otherwise low-cost, carbon-neutral, and independent European energy system designs (Methods–Candidate designs). The indicators (Fig. 1; Table 1) consider three theories of distributive justice–equality, equity, and utilitarianism–and consider three key distributional impacts–investment cost, job creation, and land use (Methods–Justice indicators). We also develop a composite indicator using public survey data to reflect a balanced and democratic approach to building a distributionally just energy system. Our results suggest that accounting for justice can help resolve system planning dilemmas, especially when leveraging public opinion to identify regionally acceptable solutions. Considering public opinion within the planning process may also lead to infrastructure capacity recommendations that are statistically different from all other candidate designs, indicating that considering justice as a design principle can help navigate the diversity of possible future energy systems. Encouragingly, we also find negligible cost-justice trade-offs in our set of candidate designs.

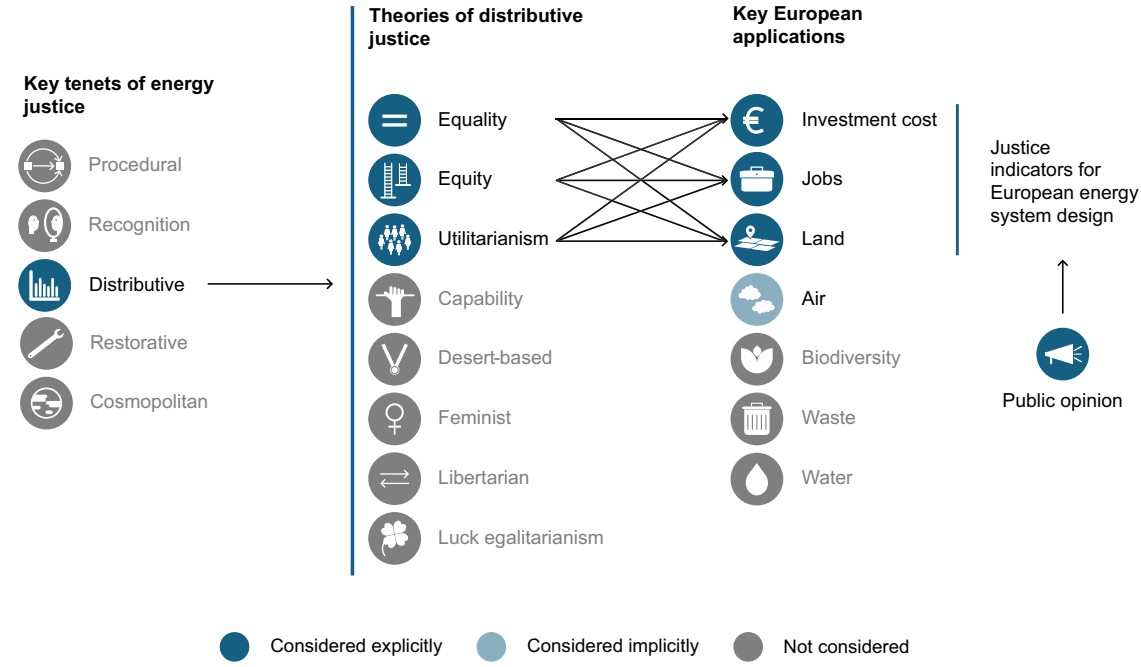

**Fig. 1 | Theoretical foundations for the nine indicators of distributive justice considered in the present study.** Here, we focus on theories of distributive justice[12] and applications that are applied in recent European energy policy and readily measurable in the context of energy systems models[16]. Applications to air impacts are implicitly considered as all candidate system designs are carbon-neutral (see "Methods"–Candidate designs). Note that the presentation of energy justice, theories of distributive justice, and potential applications are representative only; other framings, theories, and applications exist[59,98].

**Table 1 | Description of indicators and their quantification**

| Impact | Description | Unit | The best-performing candidate design… |
|---|---|---|---|
| Theory of distributive justice: Equality | | | |
| Cost | Investment cost per person | Total investment cost per person (€/person) | … minimises the sum of percent differences to equal cost per person. |
| Jobs | Jobs created per person | Total jobs per person (€/person) | … minimises the sum of percent differences to equal job creation per person. |
| Land use | Share of land capacity required to host energy technologies | Percent land use ($km^2$ needed /$km^2$ available) | … minimises the sum of percent differences to equal land use per person. |
| Theory of distributive justice: Equity | | | |
| Cost | Investment cost per person | Total investment cost (€) | … maximises the ratio of investment in priority countries to investment in non-priority countries. |
| Jobs | Jobs created | Total jobs (FTE) | … minimises the sum of percent differences between the job creation per country and the fossil job distribution per country in 2022. |
| Land use | Share of land capacity required to host energy technologies | Percent land use ($km^2$ needed /$km^2$ available) | … minimises the sum of percent differences between land use per country and all the available land per country after considering additional protections required to meet land protection goals. |
| Theory of distributive justice: Utility | | | |
| Cost | Total investment cost | Total investment cost (€) | … minimises investment cost. |
| Jobs | Total job creation | Total jobs (FTE) | … maximises job creation. |
| Land use | Total land use requirements | Total area ($km^2$) | … minimises land use. |

The performance of candidate designs is assessed based on how well the design fulfils its goal, as tracked via a minimisation or maximisation (see right-most column). Job creation in full-time equivalent jobs (FTE) is calculated for six strategic technologies: solar PV, onshore wind, offshore wind, batteries, electrolysers, and heat pumps (Methods–Calculating system impacts)[73]. The priority countries for the Equity–Cost indicator are those with 90% or less of the GNI per person in the set of all countries considered in the analysis, as presented in the Cohesion Fund[74]. The land use protections considered for the Equity–Land indicator refer to the need to protect 30% of land mass, as stipulated by the European Union Biodiversity Strategy[73]

## Results

### Ideal infrastructure designs according to single justice indicators

To identify just infrastructure designs, we develop a set of nine indicators for distributive justice. The indicators are based on three theories of distributive justice and three locally relevant impact categories (Fig. 1). We consider equality, equity, and utilitarian theories of distributive justice, which are well-established and can be readily applied to a technical modelling study (see "Methods"–Justice indicators). In brief, an equality theory argues for an even (equal) distribution of benefits and burdens, an equity theory argues for distributions of benefits and burdens to rectify pre-existing differences among parties, and a utilitarian theory argues for distributions of benefits and burdens that maximise the overall welfare, or utility[12]. As impact categories, we consider investment cost, job creation, and land use (see "Methods"–Impact categories). The value of these indicators is providing a mathematical interpretation of theory and locally relevant impact categories (Table 1), which can then be used to compare different candidate designs.

The best-performing candidate designs have the highest indicator scores considering aggregate performance over all countries (Methods–Ranking candidate designs). The best-performing candidate designs according to the equity and equality indicators minimise the total deviation from perfectly equal or equitable distributions. Our utility-based indicators measure the total impacts (summed across the study area; Methods–Utilitarian indicators), the overall equality-based indicators measure how evenly impacts are distributed proportionally to population and usable land (Methods–Equality indicators), and the equity-based indicators measure how well impacts are distributed according to existing EU equity schemes targeting climate investment, green job creation, and land use impacts (Methods–Equity indicators). Specifically, the cost-equity indicator aims to channel investment towards countries with lower gross national income (GNI) per capita, the jobs-equity indicator aims to create jobs according to the current distribution of fossil fuel workers, and the land-equity indicator aims to ensure that all countries can meet land protection targets. These equity schemes align with those in the EU Cohesion Fund, the Just Transition Mechanism, and the EU Biodiversity Strategy, respectively.

Throughout, we focus on the impacts stemming from six key technologies[35]: solar photovoltaics (PV), onshore wind, offshore wind, batteries, electrolysers, and heat pumps (Methods–Calculating system impacts).

We find that decision-makers using justice as an *ex-post* criterion for selecting an infrastructure design may struggle to identify solutions that meet distributive justice goals. Of the hundreds of candidate designs considered, no design exactly matches the ideal distributions, as shown by the gaps between the target and the best-matching distributions in Fig. 2. The fact that an exact match is not identified for any of the nine indicators suggests that achieving energy system designs that fulfil the "ideal" solutions requires integrating justice objectives within the design development (e.g., within an optimisation model[15]) rather than relying on candidate designs generated with other objectives in mind. The benefit of doing so appears to be more pertinent for some indicators than others: for instance, no candidate design only invests in countries with lower GNI per capita, which is required according to the cost-equity indicator.

Selecting system designs to be distributionally just may still entail a problematic distribution of impacts. For example, the candidate design that provides the lowest overall land use for all of Europe uses a far higher proportion of the available land use in the United Kingdom (UK) than in other countries. Likewise, the candidate designs that best support equal job creation and equal land use create disproportionate job benefits in Iceland (IS) and land use requirements in Lithuania (LU). Even though these impacts are relatively small in absolute terms (see Supplementary Fig. 1 and Supplementary Fig. 2) and results are driven by fundamental system properties, like demand patterns and resource availabilities, deviations from the ideal distributions may be construed as unfair. The risk of perceived unfairness may be heightened when one country carries a significant burden or realises greater benefits. One approach to further equalise benefit and burden sharing and potentially increase political acceptability would be to seek out candidate designs that minimise the maximum deviations from ideal distributions. Supplementary Fig. 3 demonstrates that considering maximum deviation increases the evenness of impact distribution, it may also entail a higher overall burden, as is the case for land use.

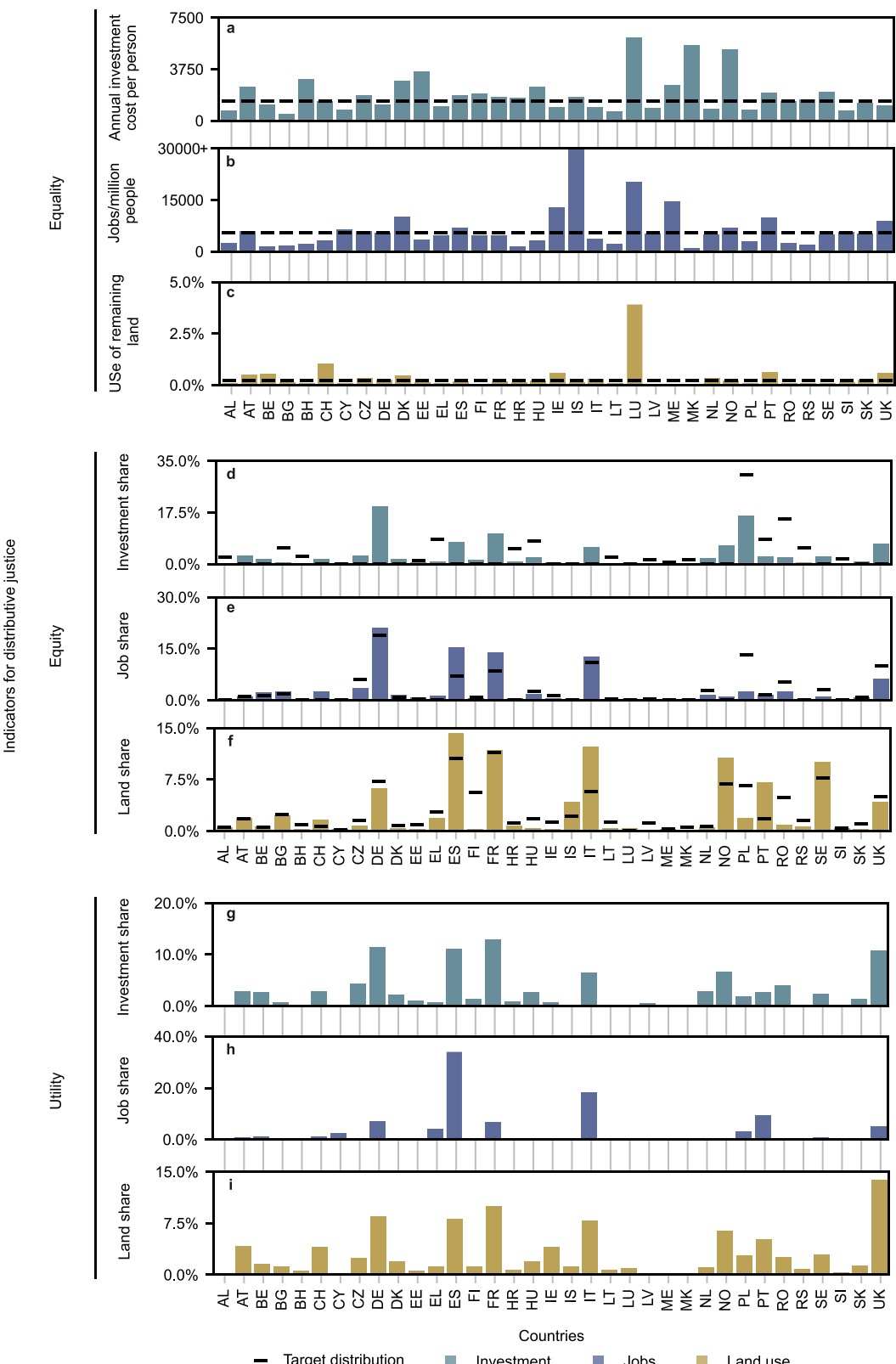

**Fig. 2 | Impact distribution for the best-performing candidate designs for each distributive justice indicator.** Each panel (**a**–**i**) represents an indicator as described by theory of distributive justice and target impact category. The solid bars display the distribution of the target impact category across case study countries (Methods–Candidate designs) resulting from the candidate design that best fulfils the corresponding justice indicator. Different bar colours indicate the different impact categories targeted by each indicator. The black lines indicate the target distributions for the equality (**a**–**c**) and equity indicators (**d**–**f**). The targets for the equality indicators are calculated per candidate design considering the total cost, job creation, and land use, respectively (see Methods–Equality indicators and Methods–Equity indicators).

**Table 2 | Trade-offs between top-performing candidate designs across impact categories**

| | Additional investment cost | Additional job creation | Additional land use |
|---|---|---|---|
| Theory of distributive justice: Equality | | | |
| Investment cost | 1% | 4% | 6% |
| Job creation | 4% | −21% | −23% |
| Land use | 4% | −50% | −57% |
| Theory of distributive justice: Equity | | | |
| Investment cost | 1% | 11% | 17% |
| Job creation | 0% | 0% | 0% |
| Land use | 9% | 0% | −3% |
| Theory of distributive justice: Utility | | | |
| Investment cost | - | - | - |
| Job creation | 5% | 123% | 140% |
| Land use | 4% | −50% | −57% |

Each row refers to one specific candidate design. The minimum cost solution (utility–investment) is used as a reference. Note that the design minimising land use also best supports equal land use. Hence, the impact trade-offs shown in this table are identical. Likewise, the lowest cost design supports the most equitable job distribution, thus the 0% trade-offs.

Designing an energy system based on a single theory of justice and impact category risks introducing trade-offs between system costs, jobs, and land use. However, our results suggest that the extent of these impact trade-offs vary.

We observe clear trade-offs in investment costs, job creation, and land use requirements between the top-performing candidate designs (Table 2). Compared to the least-cost option, candidate designs targeting different impact categories and prioritising different visions of distributive justice can lead to diverse gross impacts, with job creation potential ranging from a 50% decrease to a 123% increase over the least-cost solution. Land use requirements range by similarly large margins, spanning from a 57% decrease to a 140% increase depending on the candidate design. Trade-offs are less prominent in terms of cost: although all candidate infrastructure designs cost no more than 10% of the lowest-possible cost solution by construction (Methods–Candidate designs), aiming specifically for equitable land use entails only 5% additional investment compared to the least-cost solution. Aiming for equal and equitable investment likewise only entails 1% of extra cost. However, pursuing one vision of justice does not necessarily come at the expense of another. The objective of minimising land use aligns with the objective of even land use. Similarly, the least-cost candidate design is also the candidate design that provides the most equitable job distribution.

Considering the broader set of all candidate designs demonstrates reduced justice trade-offs as compared to the per-indicator top-performing candidate designs. Across the entire set of candidate designs, positive linear correlations exist for one-half of all tested indicator and impact pairs, indicating complementarity between these visions of justice (Supplementary Fig. 4). Minimising land use requirements is the most conflicted objective, the fulfilment of which works against job creation and shows negative correlation with almost all other impact categories (Supplementary Fig. 4 and Supplementary Fig. 5). The reason behind this relationship is in how PV is installed: candidate designs with large shares of open field PV are both job- and land-intensive compared to other technologies. Mitigating this trade-off while maintaining the same PV capacity requires either radically increasing the energy production density of open-field PV or hosting PV on top of existing infrastructure, like façades (Supplementary Fig. 6). Results for equality-based designs are less conflicting than designs targeting equity- and utility-focused designs. Negative correlations between different justice indicators occur in 33% of pairings (Supplementary Fig. 4).

Critically, including justice as a decision-making criterion does not necessarily entail significant cost increases. Lesser trade-offs are associated with cost: the cost-utility indicator has lower absolute average and maximum linear correlations than other indicators (Supplementary Fig. 4). We observe complementarity between lower-cost and higher-justice candidate designs for four indicators: equitable and equal distribution of investment and job benefits (Fig. 3). The only design objective that conclusively results in cost increases is maximising job creation, while no relationship can be established at the 95% confidence level for the remaining three indicators.

**Refining the design space for practical implementation**

Our results highlight the potential of using justice indicators to identify justice-aligned candidate designs and the theoretical alignment between different visions of justice. However, practical implementation requires decision-makers to reconcile the infrastructure recommendations suggested by one vision of justice versus another: 43% of candidate designs are ranked in the top 10% of options according to at least one indicator (Supplementary Fig. 7). Therefore, we next investigate the infrastructure recommendations in terms of installed technology capacity to test whether applying justice as a design criterion can support robust infrastructure development targets (Methods–Sensitivity analysis). We present design space reduction in terms of the range of installed technology capacity suggested by the top 1% of designs (4 candidate designs) compared to the potential design space of the entire set of candidate designs. A range reduction of 100% indicates that applying a justice criterion leads to a unique capacity recommendation. Conversely, a range reduction of 0% indicates no decrease in possible capacity ranges compared to the original set of candidate designs.

Fig. 4a shows the reductions in design space for key technologies according to the top 1% performing designs for each of the nine indicators. Applying the justice criteria reduces the design space by a median of 83% (74% in terms of averages). These median benefits also exceed the reduction potential of randomly drawn candidate designs by 14% (and up to 11% when considering averages; see Methods–Sensitivity analysis). These results demonstrate a clear advantage of considering justice as a design criterion, particularly given the marginal additional effort to identify the most just designs compared to the initial effort required to generate the set of all candidate designs.

Seven out of nine indicators have mean range reductions of 74% or higher and eight out of nine indicate design space reductions of 80% or higher for at least two technologies. The outlier indicator is for maximising jobs (utility-jobs), which presents a far lower range reduction potential, with a median reduction potential of 39% (mean reduction 40%). The relative diversity of options for creating many jobs (Supplementary Fig. 8) explains the relatively poor performance of this indicator. So, although aiming to maximise job creation appears less helpful than other indicators in reducing design space, it also highlights the greater flexibility available to policymakers in determining which of the top-performing candidate designs is most politically feasible.

Notably, equality and equity principles exhibit greater potential benefits in reducing the design space than a utilitarian approach. Aiming to provide equal or equitable impact distribution provides a median design-space reduction of 91% compared to only 65% for the utilitarian approach. Equality-based indicators also retain their comparatively stronger performance when considering a more relaxed threshold for what is "most just" (see Supplementary Table 2). In other words, adopting equality of investment as the guiding principle for distributive justice provides the most practical benefit in terms of identifying the most refined set of technology capacity targets.

The overall benefit of narrowing the design space decreases as more candidate designs are considered "top" performing designs.

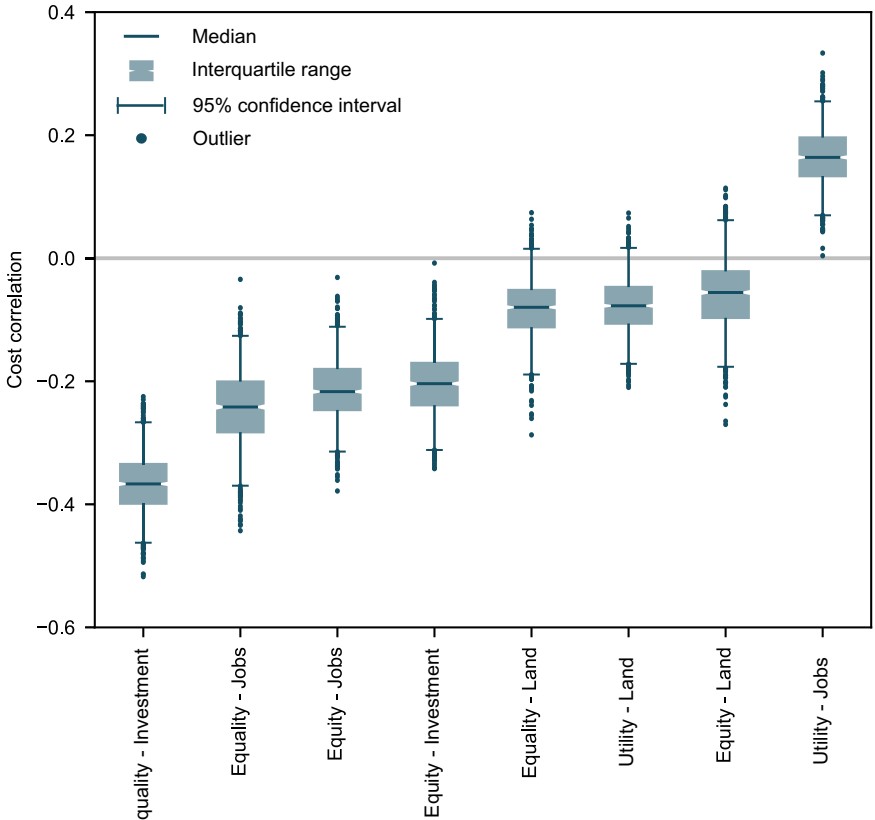

**Fig. 3 | Cost-justice trade-offs across all candidate designs (*n* = 353).** The box-plots show the robust linear correlation coefficient for 1000 bootstrapped data samples. Centre line, median; box limits, upper and lower quartiles; whiskers, 95% of all data; points, outliers. The sign of the correlation cannot be determined where the 95% interval includes 0.

Increasing the share of tracked candidate solutions from 1% to 5% and 10% (4, 18, and 35 candidate designs, respectively) decreases the median range reduction potential from 83% to 61% to 45% (average range reduction potential: 74% to 59% to 47%). Performance decreases are inconsistent across distributive justice theory and impact category: total performance ranges change from only 1% to 50% across indicators (average: 27%; see Supplementary Table 2). These results show that the benefits of using distributive justice to support design-space reduction decrease as an increasingly lower threshold for a "just design" is considered.

For solar PV, considering justice as an additional design criterion always results in a higher-than-average design-space reduction. The reduction is less consistent for other technologies. The uncertainty surrounding heat pump capacity requirements can be almost entirely resolved with one indicator (equality-investment) and only weakly resolved with another (equity-land use). Some of this variability is attributable to the absolute capacity ranges for each technology. For example, the range of possible solar PV capacity is 6.3 terawatts (TW) while the range of heat pump capacity is only 1.6 TW.

The recommended capacity ranges help policymakers refine their energy transition targets and understand the magnitude of the required system transformation (Fig. 5a). In particular, the results indicate that electrolyser capacity must increase several hundred times the current capacity[36] to achieve net-zero and distributionally just sector-coupled Europe. It is also possible to prioritise some technologies over others: onshore wind emerges as the most important technology in terms of capacity requirements according to eight of the nine indicators. This information can support policymakers in prioritising European funding and support schemes.

The value of planning support in determining absolute targets and technology prioritisation varies by indicator and technology. Even

though the ranges suggested by Fig. 5 are far smaller than the range of technical possibilities, the estimated capacity ranges per technology all vary by hundreds of gigawatts (GW) for at least one indicator. Planning benefits at the European scale also do not necessarily transfer to a sub-European scale: the distribution of total European capacity can vary significantly from country to country even when total technology capacity ranges are well-defined (Supplementary Fig. 8–16). For example, although the top 1% of minimal investment candidate designs have comparatively high agreement about how much offshore wind is required (Fig. 5a), each of the top candidate designs suggests very different national technology targets for France, Germany, Norway, and the United Kingdom (Supplementary Fig. 16).

Fig 5a reveals two other practical considerations. First, achieving distributional justice in practice may depend on procedurally just processes. Here, onshore wind almost uniformly emerges as the most important technology in terms of absolute capacity for achieving a distributionally just energy system when considering the top 1% of most just candidate designs. However, onshore wind is an important technology across all candidate designs (Supplementary Fig. 17). As such, installing more onshore wind does not necessarily lead to more just outcomes (Supplementary Fig. 18). Instead, realising the distributional benefits associated with onshore wind requires clearly defining local benefits and public participation in the planning process[37].

Second, the candidate designs are all technically feasible, but the target installation ranges in Fig. 5a show that the designs are not necessarily compatible with ongoing trends. Across the nine indicators, there are highly just candidate designs that recommend not installing particular technologies (zero capacity). This situation emerges for solar PV (minimum and equal land use), offshore wind (equitable investment and land use), and heat pumps (maximal job

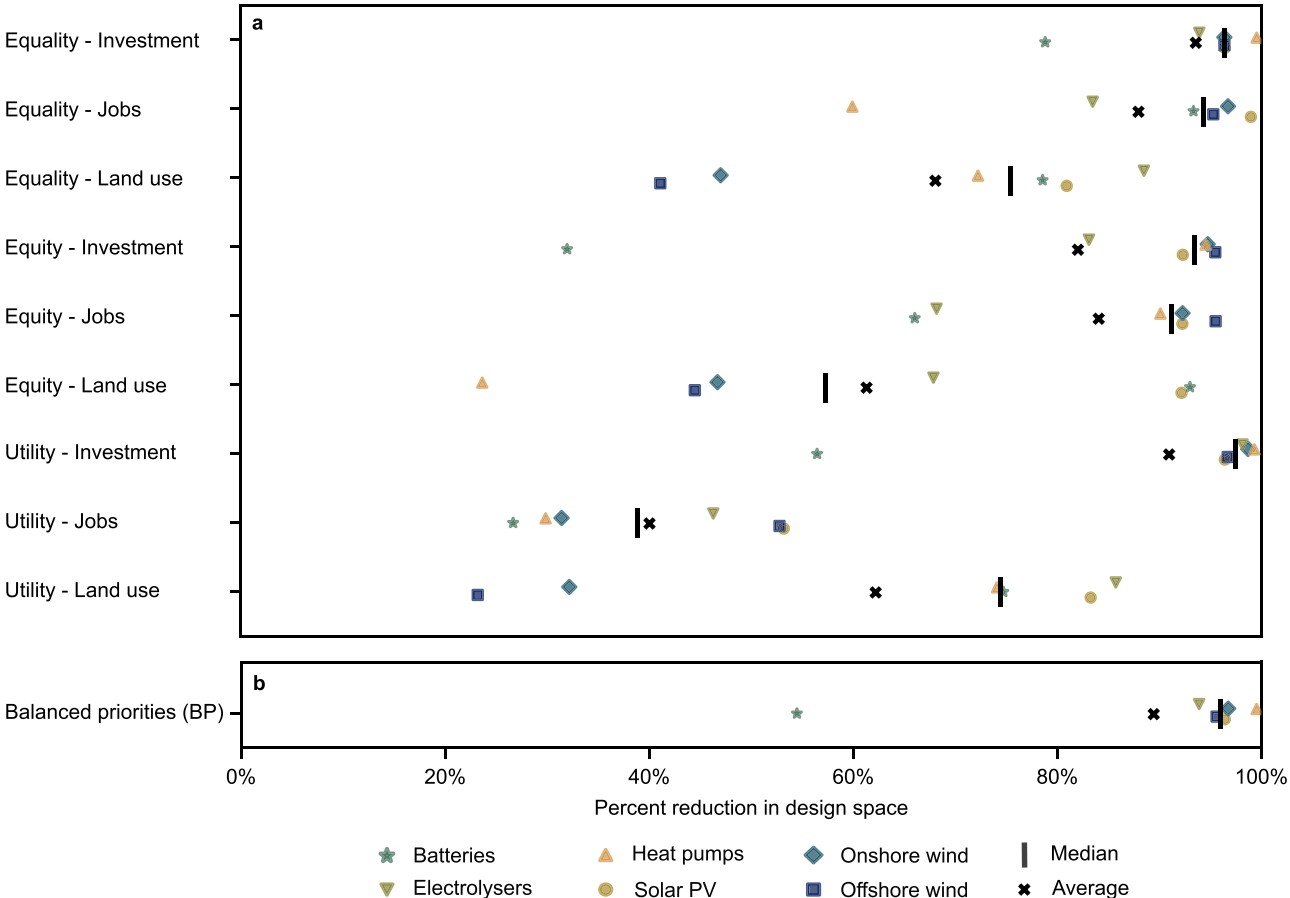

**Fig. 4 | Design space reduction by technology. Each symbol corresponds to a key technology.** Black lines and black x's indicate the median and average percent reductions across the six technologies considered per indicator, respectively.

Results shown for (**a**) each of the nine justice indicators for distributive justice and (**b**) the balanced priorities (BP) approach for the top 1% candidate designs (4 designs).

creation). However, the EU has clearly stated their ambitions to be a leader in these technologies[35], and Europe already has 263 GW of solar PV[38], 30 GW of offshore wind capacity[39], installed nearly 30 GW of heat pump capacity in 2022 alone[40], and deployed nearly 2.2 GW (3.3 GWh) of new utility-scale battery storage in 2022[41].

The mismatch between the actual installed capacities and the low-capacity model-based designs raises a key point: Although a candidate design might be just and technically feasible, it is not necessarily practical. In the European case, it seems unlikely that low-carbon technologies would be decommissioned to achieve a specific system design, no matter how just it might be. For justice to be considered a practical design criterion, some aspects of the current system should be considered, i.e., planners should consider a brownfield rather than a greenfield approach. The minimum capacity installation recommendations for electrolysers and onshore wind are 300 GW and 675 GW; policymakers can robustly target these installation levels as supporting just energy systems on the basis of any of the nine justice indicators.

### Accounting for public opinion in infrastructure planning

Pursuing a system defined by any single indicator for distributive justice would overlook the variety of interpretations for a just transition and potentially be politically infeasible in a democratic European setting. To consider a more representative and democratic interpretation of distributive justice, we also define a composite justice indicator to reflect the "balanced priorities" (BP) of the European population. The indicator is calculated using the preference-weighted average of the nine indicators (Supplementary Table 1). We extract popular

preferences from three Eurobarometer opinion surveys conducted by the European Union between 2019 and 2024[26,42,43] (Methods–Balanced priorities and Supplementary Tables 3-5). The candidate designs that best meet the BP indicator are considered most just for the European population. We compare the installation ranges and investment costs suggested by the BP indicator to those suggested by single indicators, and test whether these suggestions differ between BP candidate designs and all other designs (Methods–Testing for differences). We also conduct a sensitivity analysis of the main results to test the robustness of our findings considering variable public preferences (Methods–Sensitivity analysis).

Notably, this approach assumes that fulfilling popular preferences delivers a distributionally just energy system. However, it could be argued that the population is not suitably informed to know what is most desirable–as may be the case in understanding current water stressors within the European Union[43]–or that self-reported preferences are the result of personal individual interests rather than for justice. Aggregate statistics can also prioritise the desires of the majority ahead of the perspectives of the minority and marginalised. At a minimum, our approach of prioritising candidate system designs with public preferences supports a democratically informed view of what a just energy system should resemble.

Fig. 6 shows the key infrastructure for the top 1% of candidate designs (4 designs) that best fulfil the BP indicator. Unlike the results for individual justice indicators, the best BP candidate designs suggest that European energy systems will need significant capacities of all six key technologies. Based on the candidate infrastructure designs used

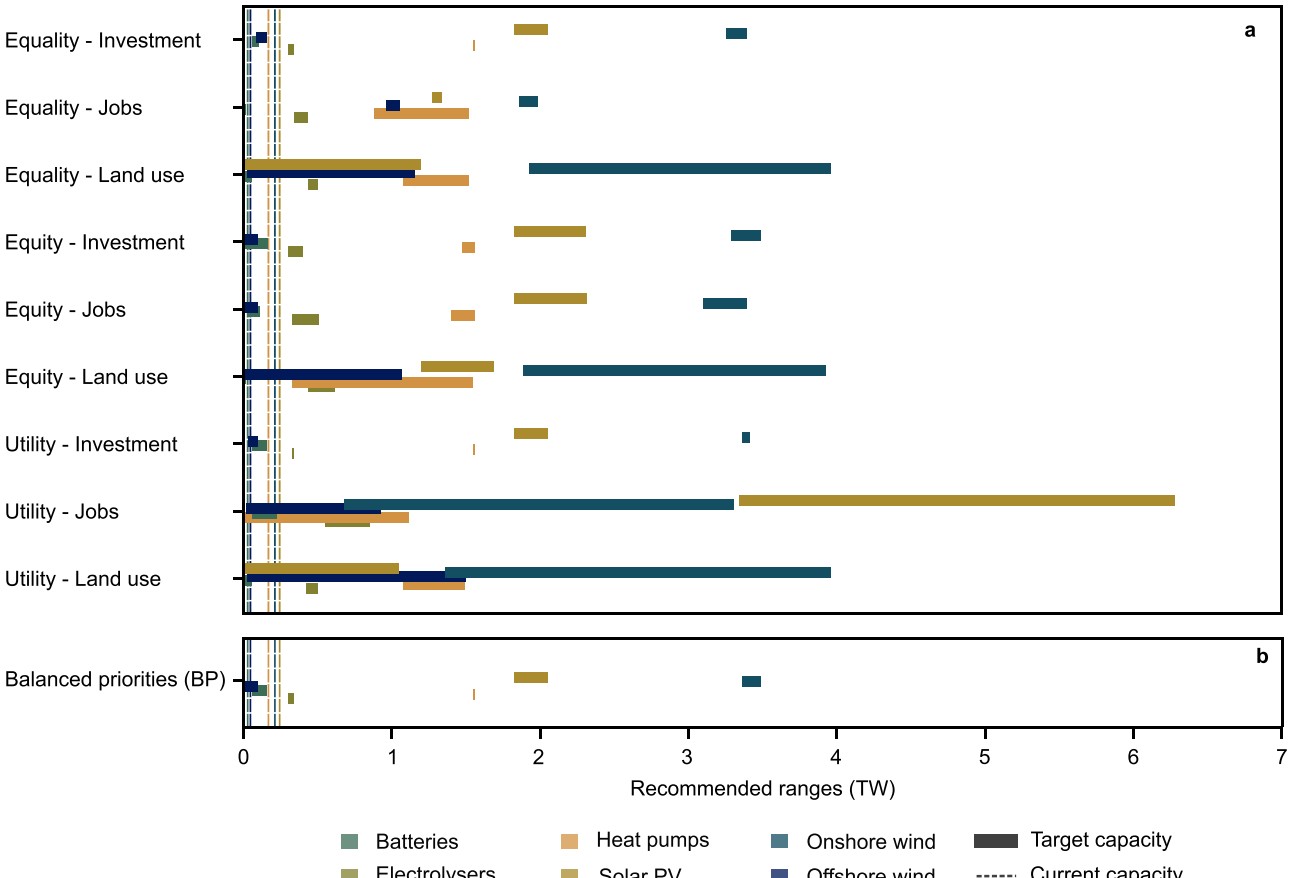

**Fig. 5 | Target capacity installation ranges of six strategic European technologies according to different approaches for distributive justice.** The dotted lines indicate current technology capacities[36,38–41] while the solid bars indicate target ranges according to the top four performing candidate designs. Results, presented in terawatts (TW), are shown for (a) the nine indicators for distributive justice and (b) the balanced priorities (BP) approach. Battery capacity is listed in power rather than in terms of energy as per the original data[19]. The current electrolyser capacity is 470 megawatts of electricity ($MW_{el}$) and is not visible in the figure.

in this study, European policymakers should aim to target in 2050 at least 60 GW of batteries, 9 GW of offshore wind, 300 $GW_{el}$ of electrolysers, 1550 GW of heat pumps, 1800 GW of solar PV, and 3400 GW of onshore wind. Except for onshore wind, these capacities all represent significant increases relative to existing installed capacities (Fig. 5b, above), one reason for which is that the candidate designs aim to support a much lower carbon and much more sector-coupled energy system than currently in existence. The BP approach also highlights the relative importance of onshore wind to achieving the most distributionally just energy system (top 1%), underlining the importance of finding socially acceptable ways of deploying the technology. As found with other interpretations of justice, installing more onshore wind does not necessarily lead to more just outcomes (Supplementary Fig. 18j), but the top 1% of candidate designs rely more heavily on onshore wind than in other cases (Supplementary Fig. 17j).

Considering a BP approach produces more consistent continental-level infrastructure development goals than considering a single justice indicator. Across all technologies, a BP approach narrows the design space by a median of 96% (mean: 89%; Fig. 4b, above). Technology-to-technology, total capacity estimates are smaller than the ranges suggested by the constituent indicators in 59% of cases (32 out of 54 technology-indicator pairs). Therefore, the BP approach presents greater benefits in reducing the design space than its constituent indicators when considered individually (Fig. 4a, above). Critically, the benefits of adopting a weighted preferences approach holds even when considering variability in public preferences (Supplementary Fig. 19 and Supplementary Table 6). These results suggest

that considering competing justice objectives is even more useful in terms of narrowing the possibilities for energy system design than considering more limited interpretations and impact categories.

The degree to which BP can help guide decision-making depends on the size of the candidate sets: the variability-reducing effect of BP decreases as more candidate designs are compared and the variability within each set of candidates increases. Relaxing the definition of the "best" candidate designs from 1% to 5%, and then 10% reduces the benefit from a median 96% design space reduction to 60% and then to 55% (average: 89% to 56% to 50%). These results are comparable to those of constituent indicators (Supplementary Table 2). These results also reinforce the idea that justice-led design choices should aim for "top-end" design choices to be more faithful to the underlying aim for distributionally just systems and to obtain the most design support.

In addition to narrowing the capacity ranges, applying a BP approach may also shift capacity targets. Across the top 10% of designs, adopting a BP approach result in statistically different absolute capacity targets at a continental level for heat pumps and batteries, and up to 30% of national level targets according to the Mann-Whitney-Wilcoxon test[44] (Methods–Testing for differences). The inconsistent link between absolute technology capacities and resulting distributional outcomes underlines that achieving a just energy transition entails much more than sampling aiming for higher technology capacities.

In line with findings for single indicators, pursuing just system designs does not imply that any major cost trade-offs will be required. In fact, the top 10% of most BP-just candidate designs are significantly

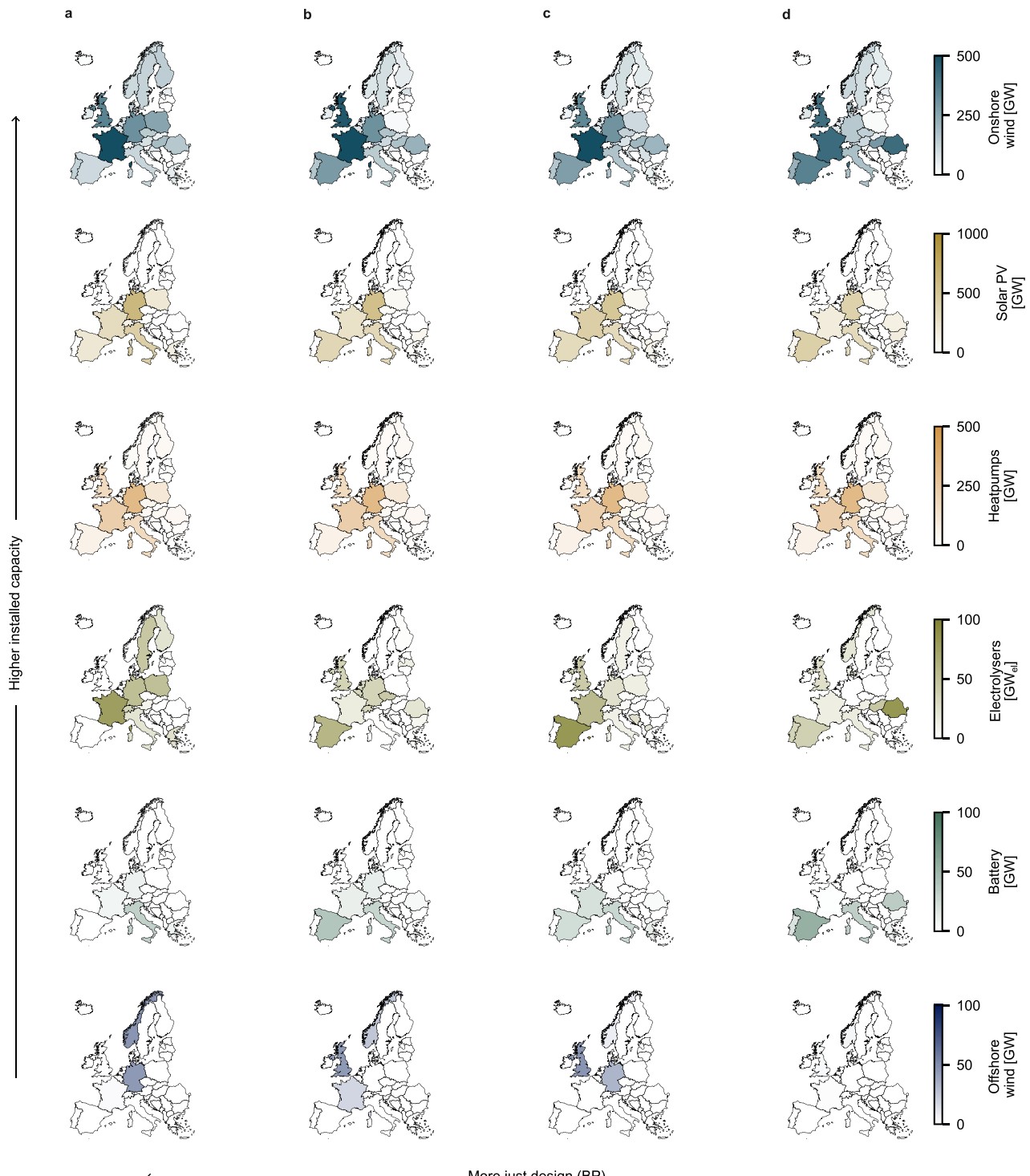

**Fig. 6 | Key infrastructure for the top 1% of candidate designs (4 designs) according to the balanced priorities (BP) approach considering European survey responses.** Columns (**a–d**) present the best-performing candidate designs ordered by rank. Column (**a**) shows the candidate design that best fulfils a balanced interpretation of distributive justice and column (**d**) shows the fourth-best-performing candidate design. Rows show national technology capacities and are ordered from most-to-least important technologies in terms of median capacity installations (Fig. 5b).

cheaper to build than the median cost of all other candidate designs (Supplementary Fig. 20). The cost of achieving the most BP-just solution results in a cost penalty of less than a 1% relative to the lowest-cost solution. The BP score across all candidate designs has a linear cost correlation of −0.70 ($p < 0.001$); the correlation is somewhat weaker when considering only the top 10% of BP solutions (−0.5, $p < 0.001$). These results reflect the comparatively high value placed on

affordability expressed by the European population (Supplementary Table 5) but overall reinforce previous results that cost is a more relevant concern towards achieving the top-most just outcomes rather than as a general challenge.

The other impacts of pursuing a BP-just design are more nuanced. The BP-optimal candidate design creates 5% more jobs and requires 7% more land than the least-cost design. However, the best BP design

creates only 53% of the jobs created by the most job-rich candidate design. Likewise, the best BP design requires 150% more land use than the least land-intensive candidate design. The fact that costs is not the most significant trade-off reflects public prioritisation of cost over the other impact categories (Supplementary Table 5). Nonetheless, these results demonstrate that the ambition of pursuing development of a distributionally just energy system is likely to produce outcomes that are unsatisfactory for at least some impact categories.

## Discussion

Legal and public consultation processes are essential for determining whether energy infrastructure development is just in a strict sense. However, justice-informed energy planning of future system-level energy infrastructure can benefit from the use of additional or alternative methods, particularly in the face of rapid and complex infrastructure development. Here, we propose that the use of energy system models incorporating justice indicators can play a role in determining the design of just energy systems.

We construct a set of nine indicators based on current European policy and apply those indicators to assess a set of technologically and spatially diverse candidate system designs. We find that pursuing justice according to narrow interpretations of justice based on single justice indicators can help narrow the set of possible candidate system designs, particularly when for indicators based on the principle of equality. Nonetheless, pursuing a narrow definition of justice can lead to significant trade-offs and retain uncertainties in required technology capacities. However, adopting a broader perspective of justice considering balanced priorities (BP) of public interests leads to results that are more reliable and representative of mixed social attitudes. Applying the BP approach helps navigate the hundreds of technologically and spatially diverse candidate designs to produce technology capacity targets that are more consistent than those produced by applying single justice indicators. The recommended ranges for capacity installations reduce the design space by an average of almost 90% across key technologies. Adopting a BP approach can also suggest statistically different capacity targets. Throughout the analysis, we found that trade-offs between investment costs and more distributionally just energy systems are limited. As such, we conclude that the policy goals of low-cost, low-carbon, and just energy systems are congruent.

Moreover, using justice as design criterion provides minimum capacity installation targets that robustly support just system design. These contributions supports the design process by reducing uncertainty about how required technology capacities, thereby streamlining policy goals and helping guide supply chain planning[35]. These conclusions hold even under multiple sensitivity tests considering the guiding justice theories and variable public sentiment.

The numerical results and best-performing candidate designs depend on several guiding assumptions. Foremost among them is the set of considered indicators and a different set of indicators would likely lead to different capacity recommendations. In addition, results depend on the number of candidate designs, the mechanism for identifying the best candidate design, the guiding equity principle, and the suitability of public survey data to balance competing priorities. However, sensitivity analyses on each of these factors show that our results remain consistent: (i) considering distributive justice is an effective measure for reducing the potential design space, (ii) considering public opinion offers a clear route for guiding infrastructure development, and (iii) the cost trade-offs to achieving more just energy systems are minor.

Our results show that accounting for justice can help reduce the field of candidate infrastructure designs, thereby providing greater certainty in infrastructure planning. However, while the motivation of this study is to understand how distributive justice can be used as a design criterion, our results help to highlight the real-world value of procedural justice. Over half of technology-country pairs show no meaningful difference between the capacity installations of the most just solutions as compared to any other candidate system design. In these cases, a given infrastructure design can usually not be said to be just or unjust on its own; instead, developing just systems depends entirely on the processes in which specific pieces of infrastructure are sourced, commissioned, operated, and decommissioned.

Likewise, our modelling results highlight a particular practical dilemma pertaining to onshore wind development. According to the top 1% of most just candidate designs for the BP indicator and eight of nine interpretations of distributive justice, onshore wind is the most important technology in terms of total capacity installations. However, developing onshore wind is fraught with challenges stemming from a lack of social acceptance. Achieving distributive justice in real terms, therefore, also requires procedurally just and socially acceptable development. Having planners engage at a local level is key for achieving both distributional and procedurally just energy systems, as the local scale is usually where opposition to energy systems investment occurs. One tangible action for reconciling system needs with local preferences is to run model-based participatory processes in communities to build problem ownership and find consensus-based actionable solutions[45]. Such a process could benefit wind development, for example, by increasing awareness about the differences between total project area versus total direct land use requirements[46,47] and finding opportunities for co-use, e.g., agriculture.

Another important aspect in moving towards a just energy system development is in characterising what is needed, in tangible terms, to move the energy transition forward. Although outside the primary focus of the present study, presenting impacts on a per capita basis (Supplementary Fig. 21) could foster discussions about how much is required to achieve a net-zero energy system and help individuals and societies reconcile their stances on the requirements of the energy transition, and what each is prepared to contribute.

Considering other types of impact categories beyond the three considered in this work (investment cost, job creation, land use) would increase the benefit of modelling activities for guiding real-world planning processes. However, more work is required to integrate further impact categories, like biodiversity, toxicity, and water impacts. Accurately characterising other impacts may require refining the representation of technologies in space and time[48]. Linking energy systems models to different types of models, such as system dynamics models[49], is one option forward. Including further impact categories also requires defining which metrics are most relevant: for industrial processes, tracking freshwater use, water stress, water ecotoxicity, and water eutrophication are all potentially pertinent metrics[50]. These challenges are most acute for immature technologies still undergoing rapid development, where resource use and local impacts are not fully characterised[48].

On the technical front, our work also points to the fact that identifying a distributionally just infrastructure design not trivial. None of the candidate designs in our set perfectly matched the ideal distributions established by our indicators, indicating a need for iterative modelling-assessment processes to identify further technically viable and socially just designs[22]. However, the value of finding better-matching solutions is not always clear, particularly within a modelling context. For example, increasing employment opportunities for current fossil fuel workers in Germany could result in a more transition-friendly political landscape; the benefit of doing so, however, eludes mathematical description. Closing this gap would require developing the associated social utility function.

External input is also valuable in forming a normative basis to define what is most just, because the mathematical formulation of indicators has a direct influence on the results[51]. Here, we base our normative indicators on dominant distributive justice principles and

current European policy to simulate a coherent legal and technical decision-making framework; relying on alternative principles and impact categories would affect which candidate designs are most attractive. Evolving public preferences would similarly affect results. Accounting for public preferences in a BP approach does enable a more democratic definition of justice; however, it does not address the procedural elements[5] related to individual infrastructure installations and, by relying on survey data, has a bias favouring the majority. As the main purpose of our study is to establish foundations for how justice principles could influence system-level design, we leave the full exploration of alternative indicators and their implications to future work. We imagine future work to consider a wider set of justice principles, impact categories (e.g., water, biodiversity, chemical leaching), supply chain impacts, and alternative normative framings[51] (e.g., Is job creation always a positive outcome?). Further work should be conducted to investigate the multi-scalar impacts of designing distributionally just energy networks and investigate the impact of decisions made to maximise system-level fairness[52] on local and regional outcomes.

In addition, while it was possible to identify which interpretations of justice, expressed by justice indicators, led to comparatively narrower ranges in estimated technology installation requirements (Figs. 3 and 4), the ranges shown in our results recall two important lessons for the modelling process. First, for a just design to be practical, it should consider the state of the current system as a starting point; some top-performing candidate designs had installed technology levels below those of the current system. Although currently operational energy assets may retire by 2050, candidate designs that effectively ban offshore wind and heat pumps are not consistent with the current state of the world. Second, modellers must be mindful of the processes used to generate candidate designs. The candidate designs considered here are extreme in that they assume Europe is energy autonomous and the manner that alternatives are generated[19]. Future work should investigate how justice behaves as a design principle for a set of less extreme candidate designs. Each of these points raise the importance of considering stakeholder opinions to help justify modelling choices and define a set of reasonable assumptions[45,53].

Finally, while our results support the utility of considering justice to navigate the diversity of candidate future infrastructure designs, several requirements must be met before justice can be applied as a design criterion in actual public system design processes. Policymakers must determine which justice principles and outcomes are most relevant to their local contexts. Identifying these factors should ideally occur through a participatory process to foster procedural justice and recognise that energy injustices occur at individual, community, and regional levels. Preferences among and between principles and outcomes should also be established, potentially by conducting dedicated surveys. "Socialising" models into existing decision-making processes[54] and communicating assumptions transparently would also help foster greater public trust and acceptance of model-informed planning processes, particularly given the sensitivity of modelling results to indicator formulation and modelling assumptions. Finally, including model-based results in court proceedings would help legitimise the use of energy models in decision-making[13], just as models of climate justice have already done[55–57].

## Methods
### Justice indicators
We derive nine distributive justice indicators based on current EU policy to assess candidate designs (Table 1). Developing quantitative indicators is an inherently normative exercise that cannot fully capture all aspects of a just transition; however, indicators also provide a transparent and replicable basis by which to compare systems, thus providing actionable input to the decision-making process[11,16]. In addition, basing the indicators on existing policy provides a normative basis for defining justice and grounds the decision-making process in existing legal norms. Most justice-informed metrics have concentrated on distributive justice[16], which is enabled by the often quantitative nature of distributional benefits and burdens, e.g., cost, jobs, and land use. Conversely, there is currently a lack of indicators and data that can describe procedural justice and recognise individual vulnerabilities and past injustices[16].

The present study focuses on distributive justice, which is an important guiding principle in energy justice and one that is suited to model-based studies[15]. We do not try to resolve the outstanding issues with indicators for other types of justice[16]. We consider three impact categories – investment cost, job creation, and land use – distributed according to three theories of distributive justice, namely equality, equity, and utility. We select these three theories as they are both well-known and readily applicable to a modelling context, utility being a predominant principle already applied in energy systems modelling (i.e., finding a least-cost design). By comparison, theories of distributive justice focused on gender, anti-racism, and capabilities[11,58] are more challenging to incorporate into a national-level modelling exercise because they require data on individuals. Considering transaction-based distributive justice is well possible within modelling exercises; however, such studies require modifying the model formulation[15], which is outside the scope of the present work.

Although we focus on a European context, the indicators and approach for their development are generic and adaptable for investigating new policy directions and other geographies. Nothing precludes applying the nine proposed indicators to other jurisdictions or scales and the development of indicators based on other impact justice principles[59]. Calculating the indicators for other geographic scopes and scales is only restricted by the existence of data and modelling results.

### Impact categories
We consider three impact categories: investment cost, job creation, and land use. We focus on these three categories for two reasons. First, these impact categories are of high policy[28,60] and public importance[26,42,43] within Europe. Second, these impact categories can be quantified with high confidence. Including a wider range of impacts, such as water use, is desirable; however, doing so first requires resolving outstanding gaps in terms of technology characterisation[50] and systems-level technology impact assessment[48] (i.e., as opposed to site-specific measurements; also see Discussion).

Generally, the data available within the context of a modelling study affects what indicators can be considered. For instance, the principle of affordability could also be proxied by the cost of electricity. In addition, the indicators focus on the distribution of costs and technologies since the model provides this information directly. On the other hand, the indicators have a lessor focus on procedural, recognition, cosmopolitan, and restorative justice[5], since the connection between the modelling results and these types of justice is more indirect. A more comprehensive assessment would consider the candidate designs' impact on specific vulnerable groups and assess infrastructure impacts associated with the entire infrastructure lifetime, including supply chain and waste disposal impacts[16].

### Calculating system impacts
To estimate employment impacts, we consider the direct operating and maintenance jobs of six strategic European technologies–batteries, direct air capture, electrolysis, heat pumps, solar photovoltaic, onshore wind turbines, and offshore wind turbines–associated as measured in employment factors (full-time equivalent jobs per capacity installed). Employment factors are taken from existing literature[61,62]. We focus on this narrower set of technologies for two reasons: first, to reflect the strategic priorities of the EU[35], and second, because it was not always entirely clear what employment factors should be used for the case of other technologies.

We consider land area impacts to be the land use required by energy technologies. We neglect technology area footprints for building technologies (e.g., rooftop solar photovoltaic, heat pumps, electric hobs) and for technologies with insufficient information was available to calculate area footprints (e.g., marine technologies, CHP biofuel extraction). We calculate technology areas requirements listed in the ecoinvent database[63] and reported by Sasse and Trutnevyte[21].

Some additional data is required to measure cost, job, and land use impacts according to equality and equity indicators. All population data is sourced from the World Bank[64]. For land use, impacts are measured per metre of available land area. Total and protected land areas are sourced from the European Land Use and Land Cover Survey[65], European Environmental Agency[66], Statistics Iceland[67], and the Joint Nature Conservation Committee[68]. Fossil fuel employment is estimated based on previous work[23] and, where needed, employment factors[69] and installed capacity[70]. Information on Gross National Income was sourced from the World Bank Group[71].

## Equality indicators
The equality indicators rank candidate designs according to how well they equally distribute positive and negative impacts across nations. For investment and job creation, impacts can be understood on a per capita basis, e.g., investment per capita and job creation per capita. For example, the best outcome for supporting equal job creation is to have job shares distributed according to national population: a country with 10% of the study area's total population should have 10% of the resulting jobs, a country with 5% of the total population should have 5% of the jobs, etc. The best outcomes for equal cost would likewise be the candidate design that best distributes investment according to national population. The best candidate design according to equal land use would be the design for which technology-related land use would be equally distributed between all countries in terms of the total available land area. In other words, energy-related land use would be equally burdensome on a percentage basis.

A target distribution is calculated per candidate design because total investment costs, job creation potential, and land use vary across candidate designs; an equal distribution depends on how much total impact there is to distribute. Candidate designs are then ranked according to how well they meet the target distribution in percent differences to avoid confounding absolute and relative differences, e.g., 10 km$^2$ of misallocated land use is more grievous to achieving equal land use in a situation where the total land use is 100 km$^2$ than in a case where the total land use is 1000 km$^2$.

## Equity indicators
Equity indicators promote differentiated benefits and burden sharing among countries in recognition of existing inequalities. The equity indicators for investment cost, job creation, and land use consequently vary in terms of the basis used to support equitable burden and benefit sharing (Table 1).

The job equity indicator is based on the Just Transition Mechanism, which seeks to support regions with high economic dependence on fossil fuel sectors[72]. We use the distribution of fossil fuel workers per country as the ideal distribution for new, low-carbon job creation: the ideal candidate provides new jobs according to the current distribution of fossil fuel workers. Here, we consider job creation from five strategic technologies[35] (Methods–Calculating system impacts).

The land equity indicator is based on the land preservation goal set in the European Union Biodiversity Strategy. Namely, all EU countries should aim to protect at least 30% of their land[73]. We identify the available land within each country, as well as the current actual status of land protection. Next, we "protect" the additional land required for each country to achieve a 30% target of land protection before seeking the candidate design that best distributes land according to the remaining available land.

The investment cost equity indicator is based on the European Union Cohesion Funding mechanism. The Cohesion Fund is the main mechanism in the EU for redressing inequities between Member States and acts by prioritising funding, including for low-carbon energy, to the Member States with Gross National Income (GNI) per person under 90% of the EU average[74]. We extend the GNI-based approach to our entire study area and prioritise infrastructure investment in the sixteen would-be recipient countries: Albania, Bulgaria, Bosnia and Herzegovina, Estonia, Greece, Croatia, Hungary, Lithuania, Latvia, North Macedonia, Montenegro, Poland, Portugal, Romania, Serbia, and Slovenia. Specifically, we maximise the ratio of infrastructure investment to these priority countries versus all other countries (Table 1). Adopting the Cohesion Mechanism as the guiding cost equity principle implies that investment into national infrastructure is a positive outcome for the ensuing effects in providing low-carbon infrastructure and access to innovation; this perspective notably differs from that in the utilitarian perspective on cost (Methods–Utilitarian indicators).

## Utilitarian indicators
The utilitarian indicators track how well candidate designs provide overall utility; in other words, the indicator tracks how well each design maximises benefits and minimises burdens of individual impact categories. In the ideal case, these indicators suggest that job creation should be maximised, while total land use and investment costs should be minimised. For the purposes of the utilitarian perspective, lower investment costs are preferable as they indicate less capital requirement for building a net-zero system. Assuming lower investment costs aligns with conventional energy systems modelling exercises that seek to identify least-cost designs and the philosophy that lower system costs translate to lower end-consumer costs[20]. There are cases where increasing investment costs may be viewed as a positive outcome, i.e. if more investment leads to higher welfare[12]. However, investigating these more complex cases is outside the scope of the present work.

## Ranking candidate designs
The candidate designs are ranked according to how well they maximise or minimise each of the indicators (Table 1). For example, the lowest-cost candidate design would be "best" according to the Utilitarian – Cost indicator, which seeks the lowest-cost solution. The best candidate designs according to equality- and equity-based indicators are those with the lowest percent differences between the proposed and reference distribution.

Focusing on the best overall match is appropriate since we consider a centralised planning problem and a European-wide score reflects energy system development across the entire geographic scope of analysis (35 countries). However, alternative formulations may be more appropriate in other contexts[16]. For example, a more equity-centric approach could consider what candidate designs lead to the lowest absolute national percent deviation compared to the ideal distribution. Adopting this "minimax" approach[75] would likely lead to different technology recommendations, for example as shown in Supplementary Fig. 3.

## Caveats to the indicators
The indicators considered here represent only one way of accounting for distributive justice in each impact category. Considering a indicators and ways of identifying best candidate designs would lead to different capacity recommendations. For example, a variant on the cost-equity indicator could be based on the Social Climate Fund[76], which allocates funding for low-carbon investment based on population at-risk of poverty rate, household emissions, households struggling to pay utility bills, total population, GNI per person, and share of reference emissions. It is an open discussion as to the most appropriate indicators for energy decision-making, and many potentially relevant indicators are yet to be identified[16].

**Table 3 | Surveys considered for creating the balanced priorities justice index**

| Survey | Fieldwork | Respondents | Countries polled |
|---|---|---|---|
| Special Eurobarometer 492: European attitudes on EU Energy Policy[26] | 09-25 May 2019 | 27 438 | EU28 |
| Special Eurobarometer 527: Fairness perceptions of the green transition[42] | 30 May-28 June 2022 | 26 395 | EU27 |
| Special Eurobarometer 550: Attitudes of Europeans towards the Environment[43] | 06 March-08 April 2024 | 26 346 | EU27 |

EU27 excludes the United Kingdom & Northern Ireland.

We do not claim that the indicators we present here are an exclusive or exhaustive representation of distributive justice or that achieving the "ideal" distribution would be sufficient to achieve all needs of a just transition, which also hinges on procedural and recognition justice, among other factors. For example, we assume that all jobs created are equally valuable, although many factors contribute to a decent work environment (e.g., gender equality, pay, etc[77,78].). We present these indicators examples of how justice indicators can help narrow the decision-making space.

In addition, cost-based indicators reflect that a just energy system considers the distribution of burdens and benefits of a low-carbon transition. However, we are limited in two main regards. First, we are unable to assess how total costs impact individual consumers due to a lack of geographically nuanced data on consumer energy burden and pricing mechanisms. Second, it is questionable whether infrastructure development costs should strictly be considered public burdens, as assumed by the centralised planning framework applied here. Investment in low-carbon technologies is associated with long-term benefits[35], and investment costs would, in any case, be shared between private and public actors[79].

**Balanced priorities**

Each indicator targets a specific and singular goal, but, in reality, multiple objectives coexist. To reflect concurrent objectives, we define an index based on the combination of individual indicators. We refer to this combination of indicators as a "balanced priorities" approach (BP). The BP index for each candidate design, $cd$, is a weighted average of how well each candidate design scores on each justice indicator, $i$. The score for candidate design $cd$ for indicator $i$ is given by $j^i_{cd}$. $j^i_{cd}$ values are normalised between 0 and 1 according to the minimum and maximum scores to produce normalising scores $\hat{j}^i_{cd}$. The purpose of normalising indicator scores between 0 and 1 is to facilitate a unit-consistent summation across indicators. The weighting factors for the justice indicators, $w_i$, are determined according to public preferences (explained next) and the weights of all justice indicators sum to 1 (Supplementary Table 2). Equation (1) gives the generic formula for the BP index calculated for each candidate design:

$$BP_{cd} = \sum_{i \,\epsilon\, \text{Justice indicators}} w_i \cdot \hat{j}^i_{cd} \qquad (1)$$

Because indicators scores are between 0 and 1 and the weighting factors sum to 1, the range of the BP index is between 0 and 1.

We determine the weights of individual justice indicators suggested by public opinion collected through Eurobarometer surveys. Since 1974, the EU Barometer survey programme has polled Europeans on political, social, and other topical issues[80]. Regular surveys are commissioned by the European Commission and the European Parliament. The surveys randomly poll at least 1000 people per country, decreasing this threshold to 500 people in countries where the total number of inhabitants is below one million inhabitants. All respondents must be 15 or older to complete the survey. The surveys are offered in the national language(s) at home, via online forms or face-to-face or telephone interviews. We review the EU Barometer database for energy- and climate-related surveys, ultimately relying on three surveys (Table 3). The surveys are conducted face-to-face, via telephone,

and online, partially as a response to COVID-19 health measures[42]. The majority of questions are formulated as multiple choice questions, allowing respondents "totally agree", "tend to agree", "agree", "disagree", "tend to disagree", "totally disagree", or give no opinion to a policy statement, e.g., "The EU must ensure access to clean energy, e.g. encourage a move away from fossil fuels towards energy sources with low greenhouse gas emissions" (e.g., QB2.2, Eurobarometer 492[26]). We focus on this type of question to obtain the percentage of people agreeing with a statement, which we use to calculate the weights for each individual indicator.

We identify a total of 18 questions among the three surveys listed in Table 3 to assign preferences to the three different distributive justice theories (equality, equity, utilitarianism) and the three impact categories (investment, jobs, and land use). We note that the surveys contain additional information not considered in this study, including preferences regarding water use and chemical exposure. This information could be integrated into future work.

The benefits of relying on Eurobarometer survey data are that the data is taken from a large, diverse set of respondents covering the majority of our study area (27 and 28 countries polled versus 35 countries considered). Leveraging pre-existing data also facilitates faster analysis than possible with a new, customised survey. The approach, however, also has some clear disadvantages. The surveys are neither fully representative nor designed to be fully aligned with our research questions. As a result, we neglect the opinions of between 5% and 18% of the population[64] considered in the modelling portion of the study (Fewer people are omitted for Special Eurobarometer 492 than for Special Barometer 527 and 550 since the United Kingdom & Northern Ireland were included in polling), risk misinterpreting survey responses, and fail to find survey questions that address each of the ten justice indicators. Aligning the modelling scope to the survey field and tailoring survey questions to the topic of just transitions would help close these gaps. Alternative forms of public engagement, e.g., a series of public workshops, could also help elicit a more precise and deeper understanding of public opinion on the development of networked energy systems. However, workshops would be difficult to conduct on a scale as large as the Eurobarometer surveys[45,81]. Relying on static survey data also introduces a limitation in that we only consider a snapshot of public priorities; however, opinions may change over time. As such, our results for what candidate designs are most distributionally should be interpreted under the assumption that views do not change throughout the study period.

**Sensitivity analysis**

Our results are sensitive to several factors, including how we identify the "most just" candidate designs and how we define the BP index. To control for these sensitivities, we perform several sensitivity analyses to ensure the robustness of our claims.

First, we test the usefulness of distributive justice indicators to reduce the design space compared sampling random designs. We randomly sample 1% of all candidate designs (4 candidate designs) and calculate the resulting range reduction per key technology. We repeat this process 1000 times and compare the average range reduction across the random draws to the range reduction achieved by using dedicated distributive justice indicators.

Second, we conduct a sensitivity analysis of the BP results by considering variability in the preferences for each impact and distributive justice theory. In the main results, the weights for the BP indicators consider the expected preference for each impact and distributive justice theory based on all relevant questions identified in the Eurobarometer surveys. In the sensitivity analysis, we sample uniformly within the range of potential preferences for each indicator and distributive justice theory (see Supplementary Table 2 and Supplementary Table 3) for constructing an alternative weighting scheme. We repeat this process 1000 times to establish the range of possible outcomes by using a BP approach, the results of which are presented in Supplementary Fig. 12.

Finally, we also check whether our claims hold with respect to the threshold value for the "most just" candidate designs. In the main results, we consider the "top-performing" candidate designs to be the top 1% of designs (4 candidate designs out of 353 possible candidate designs). We also generate results that consider the top 5% and top 10% of designs (corresponding to 18 and 35 candidate designs, respectively). Complete results are available as Supplementary Data 1.

### Testing for differences

We conduct a two-sided Mann-Whitney-Wilcoxon test[44] (also known as the Mann-Whitney U test) to assess whether the distributions in national technology installations differ significantly between the top 10% most just candidate designs and all other candidate designs. We apply the Mann-Whitney-Wilcoxon test as it is non-parametric, thus suitable to analyse the current distributions, which are frequently zero-inflated and right-skewed due to the SPORES method with which candidate designs were generated (see Methods–Candidate designs). We conduct a test with a 5% alpha level for total capacity installations and national technology capacity each country, which introduces the risk of multiple testing errors. We provide additional results applying the Holm-Bonferroni correction on our $p$-values to control for multiple testing errors[82,83].

### Candidate designs

To understand how justice can be integrated into a model-based decision-making process, we investigate the justice impacts of literature candidate infrastructure designs. The benefit of coupling a justice assessment to candidate system designs produced solely for technical, sustainability, and financial impacts is that it allows us to replicate the process of integrating justice as a design criterion into existing planning processes. Numerous candidate infrastructure designs for the European energy system can be identified in the literature[20,21,84] and could all be considered within the framework of our study. However, we consider results from Pickering et al[19]. because they apply a well-established energy system modelling framework and openly provide detailed results. 353 unique candidate designs are sourced from the sector-coupled Euro-Calliope model and generated using the Spatially Explicit Practically Optimal Results (SPORES) method[85]. The candidate designs all support an independent, carbon-neutral European energy system at no more than 10% of the least-cost alternative. The results are produced for a snapshot year, i.e. the model does not consider infrastructure transition pathways as possible in other models[84,86,87]. Although the model produces results for 98 sub-national regions, we aggregate results to a national level to mimic the focus of European Union policymaking. Thirty-five countries are considered: the EU28 except for Malta (Austria, AT; Belgium, BE; Bulgaria, BG; Croatia, HR; Cyprus, CY; Czechia, CZ; Denmark, DK; Estonia, EE; Germany, DE; Greece, EL; Finland, FI; France, FR; Hungary, HU; Ireland, IE; Italy, IT; Latvia, LV; Lithuania, LT; Luxembourg, LU; Netherlands, NL; Poland, PL; Portugal, PT; Romania, RO; Spain, ES; Slovenia, SI; Slovakia, SK; Sweden, SE; and the United Kingdom & Northern Ireland, UK), Albania (AL), Bosnia and Herzegovina (BH), Iceland (IS), North Macedonia (MK), Montenegro (ME), Norway (NO), Serbia (RS), and Switzerland (CH).

Considering energy system transition pathways and their associated climate impacts via cumulative emissions profiles would also provide a fuller picture of just climate transformation than the present approach, which only considers a static view of energy systems. Transition pathways with higher cumulative emissions contribute to higher global temperatures and climate impacts for vulnerable populations[88]; higher emissions profiles emerge, for example, when planning focuses only on near-term benefits (myopic planning) rather than considering long-term system goals[86]. Widening the scope of the analysis to assess the overall climate impacts of the energy transition would introduce an additional dimension of justice and strengthen the link between energy justice and climate justice[89].

### Reporting summary

Further information on research design is available in the Nature Portfolio Reporting Summary linked to this article.

### Data availability

The data used and generated in this study are available in the Code Ocean database[90] (https://doi.org/10.24433/CO.2884991.v2). The data underlying the figures in this study are available as Supplementary Data 1 as an electronic spreadsheet. All maps have been generated using data files from the World Food Programme[91] and modified using QGIS[92]. The colour scheme was developed by Fabio Crameri[93]. Figure 1 was created by the authors using Inkscape[94].

### Code availability

All analysis was performed using Python[95] and R[96] software on PyCharm[97] using an academic license. The custom computer code used to generate these results may be found in the linked CodeOcean repository[90] (https://doi.org/10.24433/CO.2884991.v2).

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

## Acknowledgements

We acknowledge B.P., F.L., and S.P. for making their research results openly available. We also acknowledge P.G., A.G., J.M., E.B., R.M., and E.L. for their helpful insights.

## Author contributions

K.E.L. conceptualised the study, prepared and conducted the formal analysis, and prepared the manuscript. G.S. conceptualised the study, supervised K.E.L., and edited the manuscript.

## Funding

## Competing interests

The authors declare no competing interests.
