## [Transparent Peer Review file · Nature Communications]

Considering distributive justice as a planning principle helps navigate a diversity of future energy infrastructure designs

Corresponding Author: Professor Giovanni Sansavini

Version 0:

Reviewer comments:

Reviewer #1

(Remarks to the Author)

The manuscript *_Considering justice as a planning principle helps navigate a diversity of future energy infrastructure designs_* is a timely work which engage with an important and relevant topic in energy systems planning and modelling. Previous research have highlighted both the need to include justice aspects in energy systems planning and the value of including a wider perspective on justice. Using a set of justice indicators and principles, the authors analyse a collection of near-optimal energy system designs, based on the SPORES method. Explicitly comparing different interpretations of justice, as well as including a broader perspective compared to previous studies, contributes to advancing the research field. Furthermore, the weighted combination of indicators (BP index) is a novel approach within the context of energy systems modelling. Although the work is novel, of good quality and generally suitable for publication, there are aspects that I think could be clarified or improved by more in-depth discussion.

I first list some overarching comments and then go through the individual chapters with specific comments and suggestions.

1. While the BP index is a valuable nuance in the work, you mention in the manuscript that the list of indicators is non-exhaustive. I think it would be valuable to emphasize that considering a different set of indicators would likely lead to different results. Additionally, the priorities (and survey responses) of people change over time and could be considerably different in, e.g. 2050.
2. From what I can see, you do not report on the improvement of distributional justice relative to the cost increase, for example in Figure 5a. The top-ranked candidate designs do not necessarily increase the justice of the system by the same amount, and it is therefore difficult to understand the cost-justice trade-offs. For indicator 6, it could be that the top-ranked design only marginally improve the justice, at a 1.1% cost increase. Could you either include this aspect in the results and discussion or justify its exclusion in the discussion/methods section of the manuscript?
3. Similarly, for the case of the most BP-just solution, there is *_only_* a 5.5% cost penalty relative to the lowest-cost solution, but again, this seems difficult to put into context without mentioning the justice improvement.
4. Although I do not necessarily disagree with the ten justice indicators, the connection between some of them and justice is rather vague. The system wide-indicators (minimise cost, land burden and maximise job creation), for instance, are not intuitively related to questions of distributional justice. The principles and the way you identified them in policy documents (SI Table 1) appear more as general decision-making principles, and not specifically than principles of justice. Could you incorporate such a linkage somewhere in the manuscript?

Abstract

* p.1, line 11-12: Consider reformulating to "...narrow interpretations of justice *_may lead to inconsistent_* infrastructure decisions...".

Introduction

* p.1, line 23-24: *_Even defining a "perfectly just" or ideal outcome may be insufficient to choose between two imperfectly just alternatives_*. I must admit that I do not understand what the sentence is meant to say.

* p.2, line 28-29: I would argue that quantitative models (if you are referring to energy system models) offer a method for generating a wide variety of energy system designs, which can be evaluated in terms of their distributional fairness. If you meant models in a broader sense (e.g. statistical models), that is not very apparent to me.

Results

- * p.5, line 106-108: "...no design exactly matches the ideal distribution, as shown by the gaps... in Figure 2." Are such solutions non-existent in reality, or only from the 441 candidate energy systems? What could be preventing that any model-generated candidate design exactly matches the ideal distribution? Is it the sampling of the near-optimal feasible space? The epsilon-constraint (slack)?
- * p.6, Fig. 2: Consider including a third label for the green area in the legend. Although one can figure out that blue + yellow = green, it would make the figure more understandable.
- * p.6, Fig. 2: Perhaps it is just not clear to me, but if the ideal distribution for e.g. "9. Equal land burden" is the minimal sum of percent differences (Table M1), wouldn't any horizontal line (i.e. where land area usage is the same for all countries) be ideal?
- * p.6, Fig. 2: Both indicator 6 and 10 are listed as equal cost share per person. Based on the y-axis label, I assume that indicator 10 should be Equal job share per person?
- * p.6, Fig. 2: Is there some technical constraint included for 5. Minimum land burden, or why is it not 0% for all countries (theoretically speaking)?
- * p.7, line 115-116: How do you justify looking at the top five performing energy system designs, as opposed to, let's say, three or ten? How does this influence your analysis?
- * p.7, line 122-124: There are two mentions of how much offshore wind capacity need to increase compared to today, is it 2-fold for offshore and 13.5-fold for onshore or vice versa?
- * p.9, line 158-165: Figure 4 and the associated text highlights something important. Previous research (<https://doi.org/10.1016/j.epr.2020.106690>, <https://doi.org/10.1038/nenergy.2017.124> and <https://doi.org/10.1016/j.apenergy.2019.113724>) have, to some extent, pointed to some technologies as more or less important for distributional justice. However, you show in your work that it is not that easy, and that such patterns may change when considering a broader interpretation of just distribution. I think you could highlight this as an importance contribution of your work by discussing this more in the manuscript.

Discussion

- * p.18, line 324: Potential typo "with greater higher technology"

Methods

- * p.22: There seems to be a mismatch between Figure 1 and the principles presented in Table M1. Number 3 and 4 in Figure 1 is number 9 and 10 in Table M1.

Supplementary material

- * p.44: In SI Table 2, there is a low agreement by respondents to the question on whether there should be more cooperation between EU member states, which results in a low weight in the balanced priorities. At the same time, autonomy and indicator 2 seem to be defined in terms of electricity self-sufficiency, at a national level (Table M1). Shouldn't respondents indicating that cooperation shouldn't be strengthened then result in a high weight for autonomy in the BP index?

(Remarks on code availability)

The code repository seems to include the data and scripts necessary to replicate the analysis. There is a README file describing the contents of the repository as well as usage instructions, both with the provided data and suggestions on how to tailor it to using other models.

I was able to install and run the scripts, which produced data and figures.

As I am not a data scientist, I am unable to review if the code is written in an efficient manner.

Reviewer #2

(Remarks to the Author)

This manuscript explores how considerations of justice can be used to rank, in relative terms, the performance (and potential desirability) of low-carbon transformation pathways for Europe. I find the paper interesting, well-written and with a methodology that is clear and appropriate. I only have some minor comments, which I hope the authors can reflect on and respond to.

Comments:

L55: Before heading in the different interpretations of justice, I would have expected an overview and discussion of the principles of justice that are being considered. Distributive justice, procedural justice, etc. and which dimensions of justice might not be covered (e.g., climate justice, which is more applicable in a developing country context).

L83-86: It might be valuable to reflect on whether all the transformation pathways that are considered for this analysis are fully equivalent in what they deliver in terms climate outcomes. If one could argue that the temperature outcomes implied by the transformations (and their implications outside Europe) this would reflect a clear additional dimension of justice as vulnerable populations would experience different climate impacts.

L211-222: This suggests that what people think is a proxy for what is just. It would be valuable to reflect whether this is universally true.

L382-383: Isn't this ambition a bit naïve? modellers are citizens, active in a deeply societally relevant field. being free from influence seems to be impossible. instead, one could emphasize processes that ensure assumptions are made and communicated transparently.

Small comments:

L12: Consistent with what?

L14: maybe the word "substantive" is better than the word "meaningful" here?

L21-22: This sentence suggests that the decision can be framed as a justice and ethical concern? Please improve grammar for readability.

L23: This is editorial but an answer is the response to a question, not the outcome of a task.

(Remarks on code availability)

Reviewer #3

(Remarks to the Author)

This study provides an interesting empirical exercise to examine energy planning in the context of justice. Although I appreciate the authors' intent, I think the study has too many limitations which leave me skeptical of both the research design and the findings.

First, the authors conceptualization of justice is flawed. As they readily acknowledge, the ten justice indicators that they analyze exclude principles of procedural and recognition justice and focus instead exclusively on distributive justice. This omission of two of three key principles of energy justice is not just a data limitation, but a fundamental problem that undermines the entire exercise analysis. I don't see a viable path forward with this study with these key principles excluded.

Second, the authors do not provide sufficient justification for the choice of ten indicators that they do use. They note that they come from EU policy, but it is not clear what they mean. For example, it is quite odd that environmental protection is reduced to a single impact – minimum land burden – when energy installations are associated with any number of short- and long-term environmental impacts, in addition to more social and cultural impacts such as loss of place attachment.

Third, there are several components of the analysis (the best I understand it), that would seem to re-enforce existing disparities. First, consider the set of criteria to keep things "equal", such as equal cost by GDP, equal land burden, and equal access job access. Keeping these equal does not redress existing disparities, which means it is not "just". Second, relying on majority public opinion to create weights for indicators does the same thing. In fact, the entire survey approach to weight the indicators seems fraught, given challenges with survey sampling. In addition, the national samples completely overlook the siting challenges that come with any energy installation, where what matters the most are local public attitudes.

Last, I find the analysis itself to be poorly explained. Few details are provided about the "candidate European energy system infrastructure designs" beyond that they come from other studies, and I find the numerical analysis to be difficult to follow (e.g., it is not clear how the installation figures are generated).

In sum, while I appreciate the intent of this study, the actual implementation is problematic, and therefore I think it fails to deliver on the desired objective of creating a methodology that can be practically applied by policymakers.

(Remarks on code availability)

Version 1:

Reviewer comments:

Reviewer #1

(Remarks to the Author)

Thank you for making considerable changes to the manuscript and responding to the comments in detail. I believe that with the changes and clarifications in the revised version the paper is now more coherent and clear about the assumptions and limitations of the method.

The distributional justice indicators have been re-constructed and better motivated than they were previously. A few detailed questions remain (see comments below), but I am happy with the changes made.

****Results****

* Figure 3. It appears like panel a) of the figure was not included.

* On p.11 you state that "Figure 5a reveals two other practical considerations. First, onshore wind almost uniformly emerges as the most important technology in terms of absolute capacity for achieving a distributionally just energy system."_, but can this really be deduced from this figure? It is true that onshore wind has the highest installed capacity in all but one principle, but only from the top 1% (four system designs). Can you see the same pattern across a larger span of designs, or could it be that also the worst system designs are dominated by onshore wind in terms of absolute installed capacity?

****Distributional justice indicators****

Some of the nine justice indicators are a bit difficult to understand and exactly how they are operationalised. Some further clarifications would help the reader better understand exactly they were implemented. The description of the ideal candidate design (for e.g. Equality - jobs) is useful, but only described for some of the principles. Table M1 is a helpful illustration, but could be referred to when the indicators are being described. This way the reader will know that there are more details below.

* Whereas the equality indicators for investment costs and job creation are clear from the description (per-capita based), it is difficult to interpret implementation of the Equality - land area principle from the text description. Reading between the lines and looking at Table M1, I understand it as that land-use from new technologies should be equally distributed between all countries.

* In the Equity - land area principle, you say that you identify the available land area within each country. Do you exclude settlements, infrastructure (roads, railways) and other categories, or is it only protected areas?

* For the Equity - job principle, you mention that you prioritise new infrastructure for the 16 countries. Is the ideal candidate then one that invest the maximum amount of infrastructure in those countries together? Is infrastructure development in other countries not affecting (be it positively/negatively) the evaluation?

* In the Utility - investment principle, you minimise investment costs (and also operational costs, making it the cost-optimal solution?), but for Equity - investment, you consider investments positive for the 16 selected countries? Am I misunderstanding the principles, or what is the reason for the different approaches?

(Remarks on code availability)

The code seems to run fine, just as before.

Reviewer #2

(Remarks to the Author)

The authors have constructively engaged with and responded to my review comments.

I am happy to support publication of the manuscript

(Remarks on code availability)

Reviewer #3

(Remarks to the Author)

The authors mostly ignored my comments, so all of the problems with the original manuscript remain. Because the authors did not seriously engage with my suggestions, I really don't have anything else to say about the study.

(Remarks on code availability)

Version 2:

Reviewer comments:

Reviewer #1

(Remarks to the Author)

The authors have addressed the comments, questions and suggestions they have received thoroughly and the few clarifications that I had have been resolved.

As such, I am happy with the current state of the manuscript and believe that it is suitable for publication in Nature Communications.

(Remarks on code availability)

I have reviewed the code in the initial review and after the first revision.

Manuscript: NCOMMS-24-21451

Title: Considering distributive justice as a planning principle helps navigate a diversity of future energy infrastructure designs

Response to reviewers:

Thank you very much for your constructive and positive feedback, and for your patience in receiving our modified manuscript. We are happy to resubmit our manuscript following a major revision where we have addressed all points of feedback. The major changes relate to:

- Updating the basis for individual indicators, as suggested by Reviewer #1 and Reviewer #3. This update has led to an entirely updated set of results.
- Streamlining the narrative of the manuscript, in line with comments from the editor, Reviewer #2 and Reviewer #3.
- Extending the sensitivity analysis for our results, in line with feedback from Reviewer #1.

All changes are highlighted in the marked-up manuscript, with new and modified text is listed in blue, while removed text is marked with ~~red-strikethrough~~. Although the manuscript features entirely new results and large sections of updated text, we note that the main conclusions remain unchanged.

Our point-by-point responses to the original comments are listed below, with original comments listed in black and our responses in blue. All page and line numbers correspond to the marked-up manuscript. Please note that we have numbered all points of feedback (e.g., R1.1 indicates comment #3 from Reviewer #1).

We believe the review process has greatly strengthened our work. Again, thank you very much for your constructive feedback and patience!

Kind regards,

The Authors

REVIEWER COMMENTS

Reviewer #1 (Remarks to the Author):

R1.0 The manuscript *_Considering justice as a planning principle helps navigate a diversity of future energy infrastructure designs_* is a timely work which engage with an important and relevant topic in energy systems planning and modelling. Previous research have highlighted both the need to include justice aspects in energy systems planning and the value of including a wider perspective on justice. Using a set of justice indicators and principles, the authors analyse a collection of near-optimal energy system designs, based on the SPORES method. Explicitly comparing different interpretations of justice, as well as including a broader perspective compared to previous studies, contributes to advancing the research field. Furthermore, the weighted combination of indicators (BP index) is a novel approach within the context of energy systems modelling. Although the work is novel, of good quality and generally suitable for publication, there are aspects that I think could be clarified or improved by more in-depth discussion.

I first list some overarching comments and then go through the individual chapters with specific comments and suggestions.

R1.1. While the BP index is a valuable nuance in the work, you mention in the manuscript that the list of indicators is non-exhaustive. I think it would be valuable to emphasize that considering a different set of indicators would likely lead to different results. Additionally, the priorities (and survey responses) of people change over time and could be considerably different in, e.g. 2050.

Thank you for your comment – we agree on the importance of emphasising how considering a different set of indicators and changing time preferences would likely lead to different results. To provide greater nuance, we have made the following changes:

- We make this connection explicit in the discussion by writing (page 19, lines 45-47) “The numerical results and best-performing candidate designs depend on several guiding assumptions. Foremost among them is the set of considered indicators and a different set of indicators would likely lead to different capacity recommendations.”
- We show how using a different prioritisation scheme would lead to new “best” designs in new Supplementary Figure 3 and describe these changes in-text on page 6, lines 2-6, writing, “One approach to further equalize benefit and burden sharing and potentially increase political acceptability would be to seek out candidate designs that minimise the maximum deviations to ideal distributions. Supplementary Fig. 3 demonstrates that considering maximum deviation increases the evenness of impact distribution, it may also entail a higher overall burden, as is the case for land use.”
- We comment on how adopting a different prioritisation scheme would lead to new “best” designs in Methods – Ranking candidate designs on page 25, lines 35-41, “Focusing on the best overall match is appropriate since we consider a centralised planning problem and a European-wide score reflects energy system development across the entire geographic scope of analysis (35 countries). However, alternative formulations may be more appropriate in other contexts¹. For example, a more equity-centric approach could consider what candidate designs lead to the lowest absolute national percent deviation compared to the ideal distribution. Adopting this “minimax” approach² would likely lead to different technology recommendations, for example as shown in Supplementary Fig. 3.”
- We extend the methods section to add a new subsection to methods called “Caveats to the indicators” (page 27), where we explain the limitations of the indicators we use here and write, “The indicators considered here represent only one way of accounting for distributive justice

in each impact category. Considering a indicators and ways of identifying best candidate designs would lead to different capacity recommendations. For example, a variant on the cost-equity indicator could be based on the Social Climate Fund³, which allocates funding for low-carbon investment based on population at-risk of poverty rate, household emissions, households struggling to pay utility bills, total population, GNI/person, and share of reference emissions. It is an open discussion as to the most appropriate indicators for energy decision-making, and many potentially relevant indicators are yet to be identified¹.”

- We extend the Methods – Balanced priorities section to state the limitations of the static survey approach (page 29, lines 32-35): “Relying on static survey data also introduces a limitation in that we only consider a snapshot of public priorities; however, opinions may change over time. As such, our results for what candidate designs are most distributionally just should be interpreted under the assumption that views do not change throughout the study period.”

R1.2. From what I can see, you do not report on the improvement of distributional justice relative to the cost increase, for example in Figure 5a. The top-ranked candidate designs do not necessarily increase the justice of the system by the same amount, and it is therefore difficult to understand the cost-justice trade-offs. For indicator 6, it could be that the top-ranked design only marginally improve the justice, at a 1.1% cost increase. Could you either include this aspect in the results and discussion or justify its exclusion in the discussion/methods section of the manuscript?

Thank you for your feedback. We appreciate how reporting the cost-justice trade-offs in more detail would benefit the reader. In line with our response to R1.3, we have modified the text to more fully contextualise the trade-offs and improvements in distributive justice. Namely, we:

- Restructure the results to include a specific subsection on “Congruency and trade-offs between candidate designs” (starting on page 8). Therein, we include a new Table 1 (page 8, line 20) that reports all impact trade-offs for the best-performing candidate solutions.
- Include all distributional justice versus impact trade-offs in the new Supplementary Figure 5. Full regression results are available in Supplementary Data.

R1.3. Similarly, for the case of the most BP-just solution, there is only a 5.5% cost penalty relative to the lowest-cost solution, but again, this seems difficult to put into context without mentioning the justice improvement.

Thank you very much for your comment. In line with our response to R1.2, we have modified the manuscript to more fully contextualise the trade-offs and improvements in distributional justice. For the BP indicator, we extend the results section in “Accounting for public opinion in infrastructure planning – Planning implications.” And add the new paragraph (page 18, lines 37-44):

“Other impacts of pursuing a BP-just design are more nuanced. The BP-optimal candidate design creates 5% more jobs and requires 7% more land than the least-cost design (Table 1). However, the best BP design creates only 53% of the jobs created by the most job-rich candidate design. Likewise, the best BP design requires 150% more land use than the least land-intensive candidate design. The fact that costs is not the most significant trade-off reflects public prioritisation of cost over the other impact categories (Supplementary Table 5). Nonetheless, these results demonstrate that the ambition of pursuing development of a distributionally just energy system is likely to produce outcomes that are unsatisfactory for at least some impact categories.”

R1.4. Although I do not necessarily disagree with the ten justice indicators, the connection between some of them and justice is rather vague. The system wide-indicators (minimise cost, land burden and maximise job creation), for instance, are not intuitively related to questions of distributional justice. The principles and the way you identified them in policy documents (SI Table 1) appear more as general decision-making principles, and not specifically than principles of justice. Could you incorporate such a linkage somewhere in the manuscript?

Thank you for this valuable comment. We have taken your feedback to heart and, in line with R3.2, we have completely revised the justice metrics to be based more closely on existing definitions of distributional justice and prominent impact categories present within the European Green Deal. This amendment strengthens both the philosophical and practical basis for selecting our indicators. An introduction to the indicators is available on page 3, lines 34-45 and in Figure 1 (page 4).

This new approach results in nine indicators (rather than the original ten) that consider equity, utilitarian, and equality theories of distributive justice applied to costs, jobs, and land use. While using nine indicators still faces some of the same limits as the original ten indicators (see comment R1.1 and new discussion in Methods – Caveats to the indicators on page 27), we strongly believe that this iteration has significantly improved the quality and theoretical basis of our work. Thank you again for this valuable feedback.

Abstract

R1.5 * p.1, line 11-12: Consider reformulating to "...narrow interpretations of justice _may lead to inconsistent_ infrastructure decisions...".

Thank you for the suggestion. In line with feedback in R2.5, we have adopted the wording to read, "Narrow interpretations of justice may lead to inconsistent capacity recommendations, but adopting a wider perspective of justice that considers the variety of public opinion can address this shortfall" (page 1, lines 12-13).

Introduction

R1.6* p.1, line 23-24: _Even defining a "perfectly just" or ideal outcome may be insufficient to choose between two imperfectly just alternatives_. I must admit that I do not understand what the sentence is meant to say.

Thank you for your comment. We have revised the phrasing to explain that models can be useful in determining what actions to take even when perfect options may not exist. We find this point, presented by Nobel laureate Amartya Sen in *The Idea of Justice*⁴, to be a powerful argument for using models in a justice-informed decision-making context.

- Page 2, lines 41-44: "By extension, models can help identify the design choices that best fulfil a given objective, even where an ideal solution cannot be identified. For example, a no-cost energy system is impossible, but models can identify the least-cost solution. Incorporating models into decision-making can, therefore, facilitate practical decision-making⁴."

R1.7* p.2, line 28-29: I would argue that quantitative models (if you are referring to energy system models) offer a method for generating a wide variety of energy system designs, which can be evaluated in terms of their distributional fairness. If you meant models in a broader sense (e.g. statistical models), that is not very apparent to me.

Thank you for your feedback. We have modified the text to be more precise. The text now reads as follows:

- Page 2, lines 44-48: “Considering justice within an energy systems model is advantageous since models can generate a wide variety of energy system designs and manage impact trade-offs, even in complex, uncertain decision-making scenarios^{5,6}. These candidate designs can then be systematically assessed for distributional impacts⁷⁻¹⁰”.

Results

R1.8* p.5, line 106-108: "...no design exactly matches the ideal distribution, as shown by the gaps... in Figure 2." Are such solutions non-existent in reality, or only from the 441 candidate energy systems? What could be preventing that any model-generated candidate design exactly matches the ideal distribution? Is it the sampling of the near-optimal feasible space? The epsilon-constraint (slack)?

Thank you for your question. The reason that no perfect design can be identified from among the candidate set stems from the fact that we apply justice criterion as an *ex-post* criterion to a set of existing candidate designs generated considering only cost and technology parameters (see Pickering et al.⁸). In other words, no ideal distribution is found because the original optimisation problem is not set to find a distributionally just design in the first place.

We comment on the ability of finding an exactly matching candidate designs in two places:

- Page 5, lines 35-41, we write: “The fact that an exact match is not identified for any of the nine indicators suggests that achieving energy system designs that fulfil the “ideal” solutions requires integrating justice objectives within the design development (e.g., within an optimisation model¹¹) rather than relying on candidate designs generated with other objectives in mind. The benefit of doing so appears to be more pertinent for some indicators than others: for instance, no candidate design only invests in countries with lower GNI/capita, which is required according to the cost-equity indicator.”
- Page 20, lines 45-48, we write: “On the technical front, our work also points to the fact that identifying a distributionally just infrastructure design not trivial. None of the candidate designs in our set perfectly matched the ideal distributions established by our indicators, indicating a need for iterative modelling-assessment processes to identify further technically viable and socially just designs¹².”

R1.9* p.6, Fig. 2: Consider including a third label for the green area in the legend. Although one can figure out that blue + yellow = green, it would make the figure more understandable.

Thank you for your feedback. In mind of your comment and other internal feedback, we have modified the colour scheme of Figure 2. Ideal distributions are now shown with black lines and best-matching distributions shown as coloured bars.

R1.9 * p.6, Fig. 2: Perhaps it is just not clear to me, but if the ideal distribution for e.g. "9. Equal land burden" is the minimal sum of percent differences (Table M1), wouldn't any horizontal line (i.e. where land area usage is the same for all countries) be ideal?

Thank you for your astute observation! You are correct that any horizontal line would be ideal for equality indicators; the reference value is calculated per candidate design. That is, a perfectly equal distribution for candidate design d is $Total\ land\ use_d / Total\ available\ land$.

To better support the reader, we have extended the text in two places:

- The caption of Figure 2 (page 7, lines 7-8): "The target distributions for the equality indicators are calculated per candidate design considering the total cost, job creation, and land use, respectively (Methods – Equality indicators)."
- Method – Equality indicators on page 24, lines 22-27: "A target distribution is calculated per candidate design because total investment costs, job creation potential, and land use vary across candidate designs; an equal distribution depends on how much total impact there is to distribute. Candidate designs are then ranked according to how well they meet the target distribution in percent differences to avoid confounding absolute and relative differences, e.g., 10 km² of misallocated land use is more grievous to achieving equal land use in a situation where the total land use is 100 km² than in a case where the total land use is 1000 km²."

R1.9 * p.6, Fig. 2: Both indicator 6 and 10 are listed as equal cost share per person. Based on the y-axis label, I assume that indicator 10 should be Equal job share per person?

We apologise for the error. We confirm that the updated Figure 2 features unique panel titles.

R1.10* p.6, Fig. 2: Is there some technical constraint included for 5. Minimum land burden, or why is it not 0% for all countries (theoretically speaking).

Thank you for your comment. All candidate designs required some land usage to fulfil anticipated energy demand⁸. Please also refer to our answer to R1.8 regarding how candidate designs were generated and best-matching candidate designs identified.

R1.11* p.7, line 115-116: How do you justify looking at the top five performing energy system designs, as opposed to, let's say, three or ten? How does this influence your analysis?

Thank you for your feedback – we agree that selecting five was an arbitrary choice. We have revised the work in two ways:

- We now focus on the top 1% of candidate designs, which translates to 4 unique designs in the main analysis. We believe that discussing the top 1% is a more meaningful reference than 5.
- We conduct sensitivity analysis on the range reduction potential of different justice approaches considering the top 1%, 5%, and 10% of designs. This approach is introduced in Methods – Sensitivity results on page 30, lines 11-15. The results of this test are presented in Supplementary Table 2 for the indicators and in-text for the BP approach on page 15, line 47- page 16, line 5, where we write:

"The degree to which BP can help guide decision-making depends on the size of the candidate sets: the variability-reducing effect of BP decreases as more candidate designs are

compared and the variability within each set of candidates increases. Relaxing the definition of the “best” candidate designs from 1% to 5%, and then 10% reduces the benefit from a median 96% design space reduction to 60% and then to 55% (average: 89% to 56% to 50%). These results are comparable to those of constituent indicators (Supplementary Table 2). These results also reinforce the idea that justice-led design choices should aim for “top-end” design choices to be more faithful to the underlying aim for distributionally just systems and to obtain the most design support.”

R1.12* p.7, line 122-124: There are two mentions of how much offshore wind capacity need to increase compared to today, is it 2-fold for offshore and 13.5-fold for onshore or vice versa?

We confirm that the main text no longer contains references to specific capacity. In any case, we apologise for the error and confirm that the original text should have suggested a 2-fold increase for offshore wind and a 13.5-fold increase for onshore wind.

R1.13* p.9, line 158-165: Figure 4 and the associated text highlights something important. Previous research (<https://doi.org/10.1016/j.epr.2020.106690>, <https://doi.org/10.1038/nenergy.2017.124> and <https://doi.org/10.1016/j.apenergy.2019.113724>) have, to some extent, pointed to some technologies as more or less important for distributional justice. However, you show in your work that it is not that easy, and that such patterns may change when considering a broader interpretation of just distribution. I think you could highlight this as an importance contribution of your work by discussing this more in the manuscript.

R1.14## Discussion

* p.18, line 324: Potential typo "with greater higher technology"

Thank you for your attention to detail. The text has been removed as part of the updated results.

R1.15 ## Methods

* p.22: There seems to be a mismatch between Figure 1 and the principles presented in Table M1. Number 3 and 4 in Figure 1 is number 9 and 10 in Table M1.

Thank you for raising this point. As part of the new results, the indicators and the associated Figure 1 and Table M1 have been revised.

R1.16 ## Supplementary material

* p.44: In SI Table 2, there is a low agreement by respondents to the question on whether there should be more cooperation between EU member states, which results in a low weight in the balanced priorities. At the same time, autonomy and indicator 2 seem to be defined in terms of electricity self-sufficiency, at a national level (Table M1). Shouldn't respondents indicating that cooperation shouldn't be strengthened then result in a high weight for autonomy in the BP index?

Thank you for your comment. As all indicators have been updated, the corresponding survey questions have also changed and the autonomy/cooperation aspect is now excluded.

R1.17 Reviewer #1 (Remarks on code availability):

The code repository seems to include the data and scripts necessary to replicate the analysis. There is a README file describing the contents of the repository as well as usage instructions, both with the

provided data and suggestions on how to tailor it to using other models. I was able to install and run the scripts, which produced data and figures. As I am not a data scientist, I am unable to review if the code is written in an efficient manner.

Thank you for taking the time to test out the code repository! We are happy that your tests were successful.

Reviewer #2 (Remarks to the Author):

R2.0 This manuscript explores how considerations of justice can be used to rank, in relative terms, the performance (and potential desirability) of low-carbon transformation pathways for Europe. I find the paper interesting, well-written and with a methodology that is clear and appropriate. I only have some minor comments, which I hope the authors can reflect on and respond to.

Thank you very much for your time and feedback!

R2.1 Comments:

L55: Before heading in the different interpretations of justice, I would have expected an overview and discussion of the principles of justice that are being considered. Distributive justice, procedural justice, etc. and which dimensions of justice might not be covered (e.g., climate justice, which is more applicable in a developing country context).

Thank you for your comment. We agree that further deepening our discussion of justice principles, both considered and omitted, would support the reader. Towards that goal, we have revised the introduction to provide a better overview of the principles of justice and to emphasize in on distributive justice. The new text (page 2, lines 5-30) now reads:

“The low-carbon energy transition offers multiple opportunities to create a more just society. Globally, decarbonising the energy system is key to mitigating the worst effects of climate change and protecting the world’s most vulnerable from climate impacts they did not cause¹³. Locally, a shift towards renewable energy can provide new employment, foster regional growth, and improve air quality¹⁴. The decentralised nature of renewables invites a wide set of stakeholders to participate in energy decision-making; for instance, through the installation of at-home solar photovoltaics (PV). Renewables’ wide deployment potential also provide broad choice in how to build future energy systems¹⁵ and independence from entrenched patterns of fossil fuel reliance¹⁶.

Capitalising on the opportunity to create more just societies through energy systems development can occur via several routes. For example, creating inclusive and accessible decision-making processes, such as via public consultation processes, support *procedural justice*¹⁷ and mitigate the risks of individual rights being trampled within the development process. Identifying who is affected by specific decisions supports *recognition justice* and can proactively help avoid creating new injustices¹⁸. Building supply chains where human rights are respected throughout supports *cosmopolitan justice*, while using energy systems development to correct prior injustices supports *restorative justice*¹⁹.

One unresolved issue is how to fairly allocate the many burdens and benefits associated with energy systems. On one side, new energy infrastructure provides investment opportunity, employment, and, depending on the technology and operational scheme, energy autonomy²⁰. On the other side, installing new technologies can also be costly, require greater land use, and result in job losses for the fossil fuel sector²⁰. The quest for *distributive justice* is highly policy-relevant given that many energy system impacts are tangible to everyday people, like

energy prices, land use, and employment opportunities. Other transition impacts are more personal but no less policy-relevant: for instance, changes to community sense of place and to personal identities linked to the growth²¹ and decline²² to specific industries can lead to tense political stand-offs. Philosophers have long argued about what it means to achieve distributive justice^{4,23}, but answering “What is just?” is a highly subjective question ~~task~~ whose answer might change over time, even for an individual.”

We have also made changes throughout the text to better contextualise how we interpret distributive justice alongside other interpretations and other types of justice, including procedural and climate justice. These changes include:

- We update the title to specify that we focus on distributive justice. The new title is “Considering distributive justice as a planning principle helps navigate the diversity of future energy infrastructure designs”.
- Page 23, lines 11-20 in Methods – Justice indicators, where we provide more context of the theories of distributive justice, “The present study focuses on distributive justice which is an important guiding principle in energy justice and one that is suited to model-based studies¹¹. We consider three impact categories – investment cost, job creation, and land use – distributed according to three theories of distributive justice, namely equality, equity, and utility. We select these theories as they are both well-known and readily applicable to a modelling context, utility being a predominant principle already applied in energy systems modelling (i.e., finding a least-cost design). By comparison, theories of distributive justice focused on gender, anti-racism, and capabilities^{4,24} are more challenging to incorporate into a national-level modelling exercise because they require data on individuals. Considering transaction-based distributive justice is well possible within modelling exercises; however, such studies require modifying the model formulation¹¹, which is outside the scope of the present work.”
- Page 27, lines 14-16 in Methods – Caveats to the indicators, where we write, “We do not claim that the indicators we present here are an exclusive or exhaustive representation of distributive justice or that achieving the “ideal” distribution would be sufficient to achieve all needs of a just transition, which also hinges on procedural and recognition justice, among other factors.”
- Page 13, lines 6-10 and page 20, lines 17-28, where we discuss the importance of procedural justice for achieving distributional justice in the context of wind energy.
- Page 30, line 43-page 31, line 3 in Methods – Candidate designs, where we discuss how model selection can be a relevant factor for linking energy and climate justice fields.

R2.2 L83-86: It might be valuable to reflect on whether all the transformation pathways that are considered for this analysis are fully equivalent in what they deliver in terms climate outcomes. If one could argue that the temperature outcomes implied by the transformations (and their implications outside Europe) this would reflect a clear additional dimension of justice as vulnerable populations would experience different climate impacts.

Thank you for raising this valuable point – we fully agree with its importance. We have amended the discussion to include this point on page 30, line 43-page 31, line 3:

“Considering energy system transition pathways and their associated climate impacts via cumulative emissions profiles would also provide a fuller picture of just climate transformation than the present approach, which only considers a static view of energy systems. Transition pathways with higher cumulative emissions contribute to higher global temperatures and climate impacts for vulnerable populations²⁵; higher emissions profiles emerge, for example, when planning focuses only on near-term benefits (myopic planning) rather than considering long-term system goals²⁶. Widening the scope of the analysis to assess the overall climate impacts of the energy transition would introduce an additional dimension of justice and strengthen the link between energy justice and climate justice²⁷.”

R2.3 L211-222: This suggests that what people think is a proxy for what is just. It would be valuable to reflect whether this is universally true.

Thank you for raising this point – we appreciate the value and the nuance of your comment. To fully address it, we have added a new paragraph to the beginning of the second results section on page 15, lines 11-18:

“Notably, this approach assumes that fulfilling popular preferences delivers a distributionally just energy system. However, it could be argued that the population is not suitably informed to know what is most desirable – as may be the case in understanding current water stressors within the European Union²⁸ – or that self-reported preferences are the result of personal individual interests rather than for justice. Aggregate statistics can also prioritise the desires of the majority ahead of the perspectives of the minority and marginalised. At a minimum, our approach of prioritising candidate system designs with public preferences supports a democratically informed view of what a just energy system should resemble.”

R2.4 L382-383: Isn't this ambition a bit naïve? modellers are citizens, active in a deeply societally relevant field. being free from influence seems to be impossible. instead, one could emphasize processes that ensure assumptions are made and communicated transparently.

Thank you for raising this very fair point. We have modified the text to adopt your suggested wording. The sentence now reads:

- Page 21, lines 41-44: “Given the sensitivity of energy system modelling results to indicator formulation and modelling assumptions, policymakers and modellers must work to establish trust in the modelling process, for example by “socialising” models into existing decision-making processes²⁹ and ensuring that assumptions are communicated transparently.”

R2.5 Small comments: L12: Consistent with what?

Thank you for your comment. In line with feedback in R1.5, the sentence now reads, “Narrow interpretations of justice may lead to inconsistent capacity recommendations, but adopting a wider perspective of justice that considers the variety of public opinion can address this shortfall.” (Page 1, lines 12-13).

R2.6 Small comments: L14: maybe the word “substantive” is better than the word “meaningful” here?

Thank you for the nice suggestion. We have changed the wording accordingly.

R2.7 Small comments: L21-22: This sentence suggests that the decision can be framed as a justice and ethical concern? Please improve grammar for readability.

Thank you for your comment. We were hoping to pay homage to the well-known *Nature Energy* article on energy justice and decision making by Benjamin Sovacool and colleagues, ““Energy decisions reframed as justice and ethical concerns.”³⁰. However, we see how this reference could be confusing and have removed the sentence in the updated introduction.

R2.8 Small comments: L23: This is editorial but an answer is the response to a question, not the outcome of a task.

Thank you for your keen eye. We have amended the sentence to read, “Philosophers have long argued about what it means achieve distributive justice^{4,23}, but answering ‘What is just?’ is a highly subjective question whose answer might change over time, even for an individual.” (page 2, line 28-30).

Reviewer #3 (Remarks to the Author):

R3.0 This study provides an interesting empirical exercise to examine energy planning in the context of justice. Although I appreciate the authors’ intent, I think the study has too many limitations which leave me skeptical of both the research design and the findings.

Thank you for taking the time to review our work. Notwithstanding your scepticism, we greatly appreciate your engaged input: it shows us that we could be more proactive in explaining and justifying some of our choices, particularly in a multidisciplinary context. We hope that the following explanations and revisions assuage your concerns.

R3.1 First, the authors conceptualization of justice is flawed. As they readily acknowledge, the ten justice indicators that they analyze exclude principles of procedural and recognition justice and focus instead exclusively on distributive justice. This omission of two of three key principles of energy justice is not just a data limitation, but a fundamental problem that undermines the entire exercise analysis. I don’t see a viable path forward with this study with these key principles excluded.

Thank you for your comment. We fully agree that representing other principles of justice would provide a more comprehensive overview of justice-informed energy systems planning; however, it has been established that indicators that acknowledge procedural and recognition justice are not yet readily

available¹. We describe these shortcomings on page 23, lines 8-10, “Most justice-informed metrics have concentrated on distributive justice and there is a recognised gap in indicators that can describe procedural justice and recognise individual vulnerabilities and past injustices¹.”

Nonetheless, we argue that future work incorporating a broader conceptualisation of justice. We write this on page 21, lines 15-18: “We imagine future work to consider a wider set of justice principles, impact categories (e.g., water, biodiversity, chemical leaching) supply chain impacts, and alternative normative framings³¹ (e.g., Is job creation always a uniformly positive outcome?).”

Despite the existing shortcomings in available justice indicators – described in-text on page 23, lines 4-7 and page 27 in Methods – Caveats to the indicators– we respectfully argue that our work still has merit (1) for the conceptual arguments it advances, the key aspects of which could easily be extended to consider additional metrics; and (2) given that we are considering transmission-level infrastructure, which entails significant technical and economic considerations.

R3.2 Second, the authors do not provide sufficient justification for the choice of ten indicators that they do use. They note that they come from EU policy, but it is not clear what they mean. For example, it is quite odd that environmental protection is reduced to a single impact – minimum land burden – when energy installations are associated with any number of short- and long-term environmental impacts, in addition to more social and cultural impacts such as loss of place attachment.

Thank you for your comment – we have taken it to heart. In line with this comment and that of R1.4, we have rebased our indicators to more closely reflect existing theories of distributional justice and a more defined scope of European policy. Namely, we review existing theories of justice these impact categories are of high policy^{32,33} and public importance^{28,34,35} within Europe. This updated procedure is described in-text at page 23, lines 27-32 and justification provided in Supplementary Table 1. By extension, we have provided an entirely new set of results.

We additionally appreciate the importance multi-faceted, energy-related environmental impacts and non-quantifiable impacts, like loss of place attachment. We express the importance of future work to include a wider set of impacts at the following places:

- Page 20, lines 35-44, “Considering other types of impact categories beyond the three considered in this work (investment cost, job creation, land use) would increase the benefit of modelling activities for guiding real-world planning processes. However, more work is required to integrate further impact categories, like biodiversity, toxicity, and water impacts. Accurately characterizing other impacts may require refining the representation of technologies in space and time³⁶. Linking energy systems models to different types of models, such as system dynamics models³⁷, is one option forward. Including further impact categories also requires defining which metrics are most relevant: for industrial processes, tracking freshwater use, water stress, water ecotoxicity, and water eutrophication are all potentially pertinent metrics³⁸. These challenges are most acute for immature technologies still undergoing rapid development, where resource use and local impacts are not fully characterized³⁶.”
- Page 23, lines 30-32, “Including a wider range of impacts, such as water use, is desirable; however, doing so first requires resolving outstanding gaps in terms of technology characterisation³⁸ and systems-level technology impact assessment³⁶ (i.e., as opposed to site-specific measurements; also see Discussion).”

We stress that these challenges are outstanding for the entirety of LCA community rather than a solely a shortcoming of our present work.

We newly reference loss of identity and sense of place on page 2, lines 24-28, where we write, “The quest for *distributive justice* is highly policy-relevant given that many energy system impacts are tangible to everyday people, like energy prices, land use, and employment opportunities. Other transition impacts are more personal but no less policy-relevant: for instance, changes to community sense of place and to personal identities linked to the growth²¹ and decline²² to specific industries can lead to tense political stand-offs.” We welcome ideas about how to account for loss of culture place attachment within an energy systems modelling study – we would gladly list these suggestions as options for future work.

R3.3 Third, there are several components of the analysis (the best I understand it), that would seem to re-enforce existing disparities. First, consider the set of criteria to keep things “equal”, such as equal cost by GDP, equal land burden, and equal access job access. Keeping these equal does not redress existing disparities, which means it is not “just”. Second, relying on majority public opinion to create weights for indicators does the same thing. In fact, the entire survey approach to weight the indicators seems fraught, given challenges with survey sampling. In addition, the national samples completely overlook the siting challenges that come with any energy installation, where what matters the most are local public attitudes.

Thank you for your comment. We agree that there are many complexities of building a fair energy system and acknowledge the difficulties of representing all complexities within a single framework, including our own. However, we argue that our work advances the understanding and management of these complexities and support a just transition. You rightly point out that our focus on the national level cannot address specific siting challenges. At the same time, investigating country-to-country inequalities are an important scale for European and international policy, as evidenced by country-by-country climate-related requirements and funding agreements (notably including the EU Emissions Trading System, Renewable Energy Directive, Social Cohesion Fund, and Social Climate Fund).

Our revised set of indicators— equity, equality, and welfare – are all based on established theories of justice (also in line with feedback in comments R1.4 and R3.1). Given the prevalence of these attitudes within European policy (see Supplementary Table 1), we feel that these indicators are important representations of actual European attitudes and help bring our results closer to practice, even if welfare-based indicators are not strictly built to address inequities. We believe engaging with practical discussions about fairness is valuable given that a major critique of justice-related work is that it is not always practical^{39,1.4}.

Finally, we acknowledge the limitations pertaining to the use of survey data, even that sourced from sources as reputable as the European Commission Directorate-General for Communication (which has been administering Eurobarometer surveys since 1974). To control for the variability in public opinion, we have performed an additional analysis demonstrating the sensitivity of our results to variable weights in the balanced priorities (BP) approach. The results of this test are presented in Supplementary Table 2 for the indicators and in-text for the BP approach on page 15, line 47 – page 16, line 5, where we write:

“The degree to which BP can help guide decision-making depends on the size of the candidate sets: the variability-reducing effect of BP decreases as more candidate designs are compared and the variability within each set of candidates increases. Relaxing the definition of the “best” candidate designs from 1% to 5%, and then 10% reduces the benefit from a median 96% design space reduction to 60% and then to 55% (average: 89% to 56% to 50%). These results are comparable to those of constituent indicators (Supplementary Table 2). These results also reinforce the idea that justice-led design choices should aim for “top-end” design choices to be more faithful to the underlying aim for distributionally just systems and to obtain the most design support.”

Thank you again for your feedback on these points.

R3.4 Last, I find the analysis itself to be poorly explained. Few details are provided about the “candidate European energy system infrastructure designs” beyond that they come from other studies, and I find the numerical analysis to be difficult to follow (e.g., it is not clear how the installation figures are generated).

Thank you for your feedback. We regret that your reading experience was less positive than that of Reviewer 1 and Reviewer 2 (see R1.0 and R2.0). We have reviewed the text to try to better support understanding and have taken the following actions:

- We have revised the introduction to the candidate energy system designs, writing on page 3, lines 26-33, “To do so, we assess 353 unique candidate European energy system infrastructure designs with a set of nine justice indicators (Figure 1). We develop these indicators to link the qualitative ideals of a distributionally just energy transition to technical planning decisions (Methods – Justice indicators). Specifically, we apply our nine justice indicators as an *ex-post* performance criterion to a set of candidate designs, which are generated using an optimisation-based model⁸. The analysed candidate designs are low-cost, carbon-neutral, and support an energy-independent Europe (Methods – Candidate designs). Doing so allows us to identify specific system designs that support these goals and distributive justice.”
- We have added a line explicitly referring the reader to the original source for the energy system designs⁸ in the methods (page 30, lines 32-33), writing, “we consider results from Pickering et al.⁸ because they apply a well-established energy system modelling framework and openly provide detailed results.”
- We revise the subsection title from “Model selection” to “Candidate designs” (page 30) to guide the reader to the extensive explanation about how candidate designs were sourced.
- We have extended the captions for all figures to better support the reader.

In addition, a native English speaker with no technical background has reviewed our manuscript to identify opportunities to enhance reader comprehension. Changes from this process are marked throughout the manuscript and the reader’s name will be added to the acknowledgements during the proofing stage.

R3.5 In sum, while I appreciate the intent of this study, the actual implementation is problematic, and therefore I think it fails to deliver on the desired objective of creating a methodology that can be practically applied by policymakers.

Thank you for your comment. We would first like to clarify that our main intent is to establish whether distributive justice could be a useful design criterion in the scope of currently applied planning methods. We fully recognise that several future steps are required before the approach could be applied to an actual public design process. We describe these steps on page 21, lines 34-46:

“Finally, while our results support the utility of considering justice to navigate the diversity of candidate future infrastructure designs, several requirements must be met before justice can be applied as a design criterion in actual public system design processes. Policymakers must determine which justice principles and outcomes are most relevant to their local contexts. Ideally, identifying these factors should occur through a participatory process to foster procedural justice and recognise that energy injustices occur at individual, community, and regional levels. Preferences among and between principles and outcomes should also be

established, potentially though conducting dedicated surveys. Given the sensitivity of energy system modelling results to indicator formulation and modelling assumptions, policymakers and modellers must work to establish trust in the modelling process, for example by “socialising” models into existing decision-making processes²⁹ and ensuring that assumptions are communicated transparently. Including model-based results in court proceedings would help legitimise the use of energy models in decision-making⁴⁰, just as models of climate justice have already done⁴¹⁻⁴³.”

Second, we respectfully believe that our implementation is, indeed, appropriate. The tradition of model-informed policymaking has been key in energy and climate planning since the 1970s⁵ and plays an important role in evidence-based policymaking today, including within Europe⁴⁴. Moreover, notable scholars including Prof. Benjamin Sovacool^{1,30}, Prof. Erin Baker¹, Prof. Raphael Heffron^{30,45}, Prof. David Konisky^{1,46}, and scholars from the United States National Laboratories^{46,47} have called for indicator-lead approaches precisely because they support practical decision-making in a way that qualitative analysis alone cannot. We do not argue against the utility of qualitative analysis; here, our intent is to help advance alternative ways of understanding and actualising distributive justice for energy systems.

References

1. Baker, E. *et al.* Metrics for Decision-Making in Energy Justice. *Annu. Rev. Environ. Resour.* 48, 737–760 (2023).
2. Vorob'ev, N. N. Minimax. *Encyclopedia of Mathematics* (2014).
3. European Union. *Regulation (EU) 2023/955 of the European Parliament and of the Council Establishing a Social Climate Fund and Amending Regulation (EU) 2021/1060.* (2023).
4. Sen, A. *The Idea of Justice.* (Belknap Press of Harvard Univ. Press, Cambridge, Mass, 2011).
5. Pfenninger, S., Hawkes, A. & Keirstead, J. Energy systems modeling for twenty-first century energy challenges. *Renew. Sustain. Energy Rev.* 33, 74–86 (2014).
6. Scheller, F., Wiese, F., Weinand, J. M., Dominković, D. F. & McKenna, R. An expert survey to assess the current status and future challenges of energy system analysis. *Smart Energy* 4, 100057 (2021).
7. Sasse, J.-P. & Trutnevyte, E. Regional impacts of electricity system transition in Central Europe until 2035. *Nat. Commun.* 11, 4972 (2020).
8. Pickering, B., Lombardi, F. & Pfenninger, S. Diversity of options to eliminate fossil fuels and reach carbon neutrality across the entire European energy system. *Joule* 6, 1253–1276 (2022).
9. Pedersen, T. T., Andersen, M. S., Victoria, M. & Andresen, G. B. Using Modeling All Alternatives to explore 55% decarbonization scenarios of the European electricity sector. *iScience* 106677 (2023) doi:10.1016/j.isci.2023.106677.
10. Sasse, J.-P. & Trutnevyte, E. A low-carbon electricity sector in Europe risks sustaining regional inequalities in benefits and vulnerabilities. *Nat. Commun.* 14, 2205 (2023).
11. Lonergan, K. E., Suter, N. & Sansavini, G. Energy systems modelling for just transitions. *Energy Policy* 183, 113791 (2023).
12. Trutnevyte, E. *et al.* Societal Transformations in Models for Energy and Climate Policy: The Ambitious Next Step. *One Earth* 1, 423–433 (2019).
13. IPCC. Summary for Policymakers. in *Climate Change 2022 Mitigation of Climate Change. Contribution of Working Group III to the Sixth Assessment Report of the Intergovernmental Panel on Climate Change* (eds. Shukla, P. R. *et al.*) (Cambridge University Press, Cambridge, UK and New York, NY, USA, 2022).
14. European Parliament & Council of the European Union. *Directive (EU) 2018/2001 of the European Parliament and of the Council of 11 December 2018 on the Promotion of the Use of Energy from Renewable Sources (Recast).* vol. L 328/82 128 (2018).
15. Tröndle, T., Lilliestam, J., Marelli, S. & Pfenninger, S. Trade-Offs between Geographic Scale, Cost, and Infrastructure Requirements for Fully Renewable Electricity in Europe. *Joule* 4, 1929–1948 (2020).
16. IRENA. *Geopolitics of the Energy Transformation: The Hydrogen Factor.* (International Renewable Energy Agency, Abu Dhabi, 2022).
17. McCauley, D. *et al.* Energy justice in the transition to low carbon energy systems: Exploring key themes in interdisciplinary research. *Appl. Energy* 233–234, 916–921 (2019).
18. Santos Ayllón, L. M., Jenkins, K. E. H. & Kerr, S. Justice by Design: Integrating energy justice and responsible research and innovation (RRI) to deliver just energy futures. *Energy Res. Soc. Sci.* 125, 103998 (2025).
19. Heffron, R. J. Energy justice – the triumvirate of tenets revisited and revised. *J. Energy Nat. Resour. Law* 42, 227–233 (2024).

20. Energy Systems. in *Climate Change 2022 - Mitigation of Climate Change* (ed. Intergovernmental Panel On Climate Change (Ipcc)) 613–746 (Cambridge University Press, 2023). doi:10.1017/9781009157926.008.
21. Shiradkar, S. *et al.* Can community based solar energy initiatives deliver on women’s empowerment in India? Evidence from rural Assam, Bihar, Jharkhand and Uttar Pradesh. *Energy Res. Soc. Sci.* 104, 103225 (2023).
22. Carley, S. & Konisky, D. M. The justice and equity implications of the clean energy transition. *Nat. Energy* 5, 569–577 (2020).
23. Lamont, J. & Favor, C. Distributive Justice. in *The Stanford Encyclopedia of Philosophy* (ed. Zalta, E. N.) (Metaphysics Research Lab, Stanford University, 2017).
24. Sovacool, B. K. *et al.* Pluralizing energy justice: Incorporating feminist, anti-racist, Indigenous, and postcolonial perspectives. *Energy Res. Soc. Sci.* 97, 102996 (2023).
25. IPCC. 2014: Summary for Policymakers. in *Climate Change 2014: Impacts, Adaptation, and Vulnerability. Part A: Global and Sectoral Aspects. Contribution of Working Group II to the Fifth Assessment Report of the Intergovernmental Panel on Climate Change* (eds. Field, C. B. *et al.*) 1–32 (Cambridge University Press, Cambridge, UK and New York, NY, USA, 2014).
26. Mannhardt, J., Gabrielli, P. & Sansavini, G. Understanding the vicious cycle of myopic foresight and constrained technology deployment in transforming the European energy system. *iScience* 27, 111369 (2024).
27. McCauley, D. & Heffron, R. J. Just transition: Integrating climate, energy and environmental justice. *Energy Policy* 119, 1–7 (2018).
28. European Commission. Directorate General for Environment. *Special Eurobarometer 550: Attitudes of Europeans towards the Environment*. <https://data.europa.eu/doi/10.2779/07854> (2024).
29. Barbrook-Johnson, P. *et al.* Economic modelling fit for the demands of energy decision makers. *Nat. Energy* (2024) doi:10.1038/s41560-024-01452-7.
30. Sovacool, B. K., Heffron, R. J., McCauley, D. & Goldthau, A. Energy decisions reframed as justice and ethical concerns. *Nat. Energy* 1, 16024 (2016).
31. Van Uffelen, N., Taebi, B. & Pesch, U. Revisiting the energy justice framework: Doing justice to normative uncertainties. *Renew. Sustain. Energy Rev.* 189, 113974 (2024).
32. European Commission. *Communication from the Commission to the European Parliament, the Council, the European Economic and Social Committee, the Committee of the Regions: The European Green Deal. COM(2019) 640* vol. 52019DC0640 15 (2019).
33. European Commission. *European Skills Agenda for Sustainable Competitiveness, Social Fairness and Resilience*. <https://ec.europa.eu/social/main.jsp?langId=en&catId=89&newsId=9723&furtherNews=yes#navItem-1> (2020).
34. European Commission, Directorate-Generate for Energy. *Special Eurobarometer 492: Europeans’ Attitudes on EU Energy Policy*. https://data.europa.eu/data/datasets/s2238_91_4_492_eng?locale=en (2019).
35. European Commission, Directorate-General for Employment, Social Affairs and Inclusion. *Special Eurobarometer 527: Fairness Perceptions of the Green Transition*. <https://europa.eu/eurobarometer/surveys/detail/2672> (2022).
36. Sacchi, R. *et al.* PRospective EnvironMental Impact asSEment (premise): A streamlined approach to producing databases for prospective life cycle assessment using integrated assessment models. *Renew. Sustain. Energy Rev.* 160, 112311 (2022).
37. McAvoy, S., Grant, T., Smith, C. & Bontinck, P. Combining Life Cycle Assessment and System Dynamics to improve impact assessment: A systematic review. *J. Clean. Prod.* 315, 128060 (2021).

38. Willet, J., Wetser, K., Vreeburg, J. & Rijnaarts, H. H. M. Review of methods to assess sustainability of industrial water use. *Water Resour. Ind.* 21, 100110 (2019).
39. Jenkins, K. E. H., Stephens, J. C., Reames, T. G. & Hernández, D. Towards impactful energy justice research: Transforming the power of academic engagement. *Energy Res. Soc. Sci.* 67, 101510 (2020).
40. Heffron, R. J. Applying energy justice into the energy transition. *Renew. Sustain. Energy Rev.* 156, 111936 (2022).
41. Winter, G. Armando Carvalho and Others v. EU: Invoking Human Rights and the Paris Agreement for Better Climate Protection Legislation. *Transnatl. Environ. Law* 9, 137–164 (2020).
42. Robiou du Pont, Y. & Nicholls, Z. Calculation of an emissions budget for Switzerland based on Bretschger's (2012) methodology. (2023).
43. Robiou du Pont, Y., Dugast, C. & Svensson, J. Consultation sur l'alignement de TotalEnergies avec l'objectif de limiter le réchauffement climatique à 1,5 °C (accord de Paris). (2023).
44. Süsser, D. *et al.* Model-based policymaking or policy-based modelling? How energy models and energy policy interact. *Energy Res. Soc. Sci.* 75, 101984 (2021).
45. Heffron, R. J., McCauley, D. & De Rubens, G. Z. Balancing the energy trilemma through the Energy Justice Metric. *Appl. Energy* 229, 1191–1201 (2018).
46. Crespo Montañés, C. *et al.* Enabling and centering equity and justice in clean energy transition research. *Joule* 7, 437–441 (2023).
47. Dutta, N. S. *et al.* JUST-R metrics for considering energy justice in early-stage energy research. *Joule* S2542435123000387 (2023) doi:10.1016/j.joule.2023.01.007.

Manuscript: NCOMMS-24-21451A

Title: Considering distributive justice as a planning principle helps navigate a diversity of future energy infrastructure designs

Response to reviewers:

Thank you very much for taking the time to review our revised manuscript and for your feedback.

All changes are highlighted in the marked-up manuscript, with new and modified text is listed in blue, while removed text is marked with ~~red strikethrough~~.

Our point-by-point responses to the new comments are listed below. All page and line numbers correspond to the marked-up manuscript. Please note that we have numbered all points of feedback (e.g., R1.3 indicates comment #3 from Reviewer #1).

Finally, please note that we have made the following minor changes to conform with Nature Communications editorial guidelines:

- Revised the end of the abstract such that the final sentence begins with, “We find that...”
- Updated the formatting of Tables 1 and 2
- Updated figure panelling scheme
- Removed the bullet points from the introduction introducing our research questions
- Removed secondary subheadings

Thank you again for your time and support in improving our manuscript.

Kind regards,

The Authors

REVIEWER COMMENTS

Reviewer #1 (Remarks to the Author):

R1.0 Thank you for making considerable changes to the manuscript and responding to the comments in detail. I believe that with the changes and clarifications in the revised version the paper is now more coherent and clearer about the assumptions and limitations of the method.

The distributional justice indicators have been re-constructed and better motivated than they were previously. A few detailed questions remain (see comments below), but I am happy with the changes made.

Thank you again for your time, attention to detail, and constructive feedback!

R1.1 (Results) Figure 3. It appears like panel a) of the figure was not included.

Thank you for your attention to detail. The original panel (a) was intentionally removed; the information is now presented in Table 2 (“Additional investment cost”).

R1.2 (Results) * On p.11 you state that “Figure 5a reveals two other practical considerations. First, onshore wind almost uniformly emerges as the most important technology in terms of absolute capacity for achieving a distributionally just energy system.”, but can this really be deduced from this figure? It is true that onshore wind has the highest installed capacity in all but one principle, but only from the top 1% (four system designs). Can you see the same pattern across a larger span of designs, or could it be that also the worst system designs are dominated by onshore wind in terms of absolute installed capacity?

Thank you for your question. Here, we are specifically referring to the top 1% of designs. We clarify the text to read:

- Page 15, lines 2-4: “Here, onshore wind almost uniformly emerges as the most important technology in terms of absolute capacity for achieving a distributionally just energy system when considering the top 1% of most just candidate designs.”
- Page 20, lines 24-26: “According to the top 1% of most just candidate designs for the BP indicator and eight of nine interpretations of distributive justice, onshore wind is the most important technology in terms of total capacity installations.”

Onshore wind is an important generation technology according to relative capacity installations across most candidate system designs; it is, therefore, possible that the worst/least just system designs are also dominated by onshore wind. However, the relative capacity installation of onshore wind demonstrates no obvious pattern in considering the least just system designs (yellow-shaded areas on Figure A, below). We can conclude the most just (blue-shaded areas on Figure A) candidate designs rely comparatively heavily on onshore wind, but that designs that rely heavily on onshore wind relative to other generation technologies are not inherently just. Moreover, the most just candidate designs are not necessarily those with the most wind: compare panels (i) in Figure A and Figure B (below) for the utilitarian-land use indicator, for example.

Your comment has helped us realise that the text in question was at risk of misdirecting the reader to the main point, which is the importance of procedural justice in achieving distributionally just systems and resolving distributional dilemmas. We have therefore made the following changes to the text:

- We add Figure A and Figure B to our Supplementary Information. They are presented as Supplementary Fig. 17 and Supplementary Fig. 18, respectively.
- We revise page 15, line 1-9: “Figure 5a reveals two other practical considerations. First, achieving distributional justice in practice may depend on procedurally just processes. Here, onshore wind almost uniformly emerges as the most important technology in terms of absolute capacity for achieving a distributionally just energy system when considering the top 1% of most just candidate designs. However, onshore wind is an important technology across all candidate designs (Supplementary Fig. 17). As such, installing more onshore wind does not necessarily lead to more just outcomes (Supplementary Fig. 18). Instead, realising the distributional benefits associated with onshore wind requires clearly defining local benefits and public participation in the planning process¹.”
- We revise page 18, lines 21-27: “In addition to narrowing the capacity ranges, applying a BP approach may also shift capacity targets. Across the top 10% of designs, adopting a BP approach result in statistically different absolute capacity targets at a continental level for heat pumps and batteries, and up to 30% of national level targets according to the Mann-Whitney-Wilcoxon test² (Methods – Testing for differences). The inconsistent link between absolute technology capacities and resulting distributional outcomes underlines that achieving a just energy transition entails much more than sampling aiming for higher technology capacities.”
- We revise page 20, lines 14-28: “However, while the motivation of this study is to understand how distributive justice can be used as a design criterion, our results help to highlight the real-world value of procedural justice...our modelling results highlight a particular practical dilemma pertaining to onshore wind development. According to the top 1% of most just candidate designs for the BP indicator and eight of nine interpretations of distributive justice, onshore wind is the most important technology in terms of total capacity installations. However, developing onshore wind is fraught with challenges stemming from a lack of social acceptance. Achieving distributive justice in real terms, therefore, also requires procedurally just and socially acceptable development.”

Figure A/Supplementary Figure 17. Ranked importance of onshore wind capacity versus other key technologies across the justice-ranked candidate designs. Across all system designs, onshore wind is frequently the most important technology in terms of absolute capacity installation (Rank 1). The technology's relative importance is consistent with the 1% of the most just designs (Figure 5). However, the least just designs can also entail high relative installed onshore wind capacities (e.g., **g**, **h**). As such, achieving distributionally just energy systems requires comparatively high shares of onshore wind, but high shares of onshore wind do not necessarily signal just energy systems. Also see Supplementary Fig. 18.

Figure B/Supplementary Figure 18. Onshore wind capacity distribution of top 10% most just candidate designs according to the nine indicators for distributive justice and the BP approach. There are statistically significant differences according to the Mann-Whitney-Wilcoxon test in recommended installed onshore capacity for some indicators (**a**, **b**, **d**, **e**, **g**, and **h**). See Supplementary Data for statistical results.

R1.3 (Distributional justice indicators) Some of the nine justice indicators are a bit difficult to understand and exactly how they are operationalised. Some further clarifications would help the reader better understand exactly they were implemented. The description of the ideal candidate design (for e.g. Equality - jobs) is useful, but only described for some of the principles. Table M1 is a helpful illustration, but could be referred to when the indicators are being described. This way the reader will know that there are more details below.

Thank you for your feedback. We have made several changes to help clarify the justice indicators:

- We move Table M1 to the main text, where it now appears as Table 1.
- We have added additional references to Table 1 (formerly Table M1) when we first introduce our study page 3, line 33 and explain the indicators on page 4, line 6
- We have added additional references to the methods sections on page 6, line 22 (Methods – Justice indicators), line 25 (Methods – Utilitarian indicators), line 26 (Methods – Equality indicators) and lines 28-29 (Methods – Equity indicators) to guide the reader to the detailed explanations.
- We extend the description of the equality indicators page 24, lines 30-35 to give examples of the ideal candidate design.
- We extend the description of the equity investment indicator to further describe the ideal distribution on page 25, lines 29-33
- We rephrase the explanation of the utilitarian indicators on page 26, lines 4-5 to clarify the outcomes of the ideal candidate designs.

R1.4 (Distributional justice indicators) * Whereas the equality indicators for investment costs and job creation are clear from the description (per-capita based), it is difficult to interpret implementation of the Equality - land area principle from the text description. Reading between the lines and looking at Table M1, I understand it as that land-use from new technologies should be equally distributed between all countries

Thank you for your feedback. We have modified the text in two places to clarify the land use impacts:

- Page 24, line 12, where we write, “We consider land area impacts to be the land use required by energy technologies”
- Page 24, lines 30-35, where we write, “The best candidate design according to equal land use would be the design for which technology-related land use would be equally distributed between all countries in terms of the total available land area. In other words, energy-related land use would be equally burdensome on a percentage basis.”

R1.5 (Distributional justice indicators) * In the Equity - land area principle, you say that you identify the available land area within each country. Do you exclude settlements, infrastructure (roads, railways) and other categories, or is it only protected areas?

Thank you for your question. Here, we only exclude protected areas. This choice was made following the language in the EU Biodiversity Strategy, which simply states that, “In this spirit, at least 30% of the land [...] should be protected in the EU.”³, with no stipulated considerations to built-up areas or other land categories.

Based on complementary data from the EU Land Cover Overview Survey (LUCAS), accounting for built-up artificial land would have no more than a 3% difference on needed additional protection (Table A). Other types of multi-purpose land, like cropland and forests, are accounted for within the

Common Agricultural Policy (CAP) Strategic Plans⁴; as such, we do not exclude these areas from consideration towards the goal of 30% land protection.

Table A. Protected area calculation with and without considering built-up areas. All areas given in km². Data sourced from the European Environmental Agency and the EU Land Cover Overview Survey.

Country	Terrestrial country area	Terrestrial area – built-up areas	Protected area	Difference to goal for total area protection if accounting for built-up areas
AL	28791	27949	6645	1%
AT	83944	80449	24603	2%
BE	30667	27066	4524	2%
BG	110994	108439	45465	1%
BH	51216	50326	4792	1%
CH	41289	38469	4126	1%
CY	9249	8676	3502	2%
CZ	78874	75424	17257	1%
DE	357582	330566	137704	3%
DK	43167	40220	6506	1%
EE	45326	44541	9522	0%
EL	132026	126694	45721	1%
ES	505983	487543	142152	1%
FI	337547	331919	45052	1%
FR	548936	518043	154571	2%
HR	56434	54649	21530	1%
HU	93013	89271	20688	1%
IE	69946	66988	9757	1%
IS	103000	102604	26512	0%
IT	300578	280769	64413	2%
LT	64899	63507	11534	0%
LU	2595	2402	984.3	3%
LV	64586	63463	11722.7	0%
ME	13880	13608	3132.1	0%
MK	25434	24967	6681.8	1%
NL	37401	32690	8500.7	3%
NO	323380	309594	100437.3	1%
PL	311928	300695	123490.8	1%
PT	91888	86181	20560.5	2%
RO	238369	231579	55900.9	1%
RS	77485	74548	8408.9	0%
SE	449718	441730	67126.3	0%
SI	20267	19387	8211.6	1%
SK	49026	47347	18321.3	2%
UK	244381	228655	67900	2%

R1.6 (Distributional justice indicators) * For the Equity - job principle, you mention that you prioritise new infrastructure for the 16 countries. Is the ideal candidate then one that invest the maximum amount of infrastructure in those countries together? Is infrastructure development in other countries not affecting (be it positively/negatively) the evaluation?

Thank you for your comment. For the Equity – job principle, we prioritise job creation to match the distribution of fossil fuel jobs in 2022 (Table 1). For the Equity – Cost principle, we prioritise infrastructure spending in sixteen priority countries following the principles set out in for Cohesion Funding⁵ (Table 1).

To support the reader, we have added additional text to explain the Equity – Cost principle on page 25, lines 29-30, including a reference to Table 1: “Specifically, we maximise the ratio of infrastructure investment to these priority countries versus all other countries (Table 1).”

R1.7 (Distributional justice indicators) * In the Utility - investment principle, you minimise investment costs (and also operational costs, making it the cost-optimal solution?), but for Equity - investment, you consider investments positive for the 16 selected countries? Am I misunderstanding the principles, or what is the reason for the different approaches?

Thank you for your comment and attention to detail; we do indeed consider slightly different approaches for these two indicators. The utility-cost indicator looks to identify a least-cost solution as consistent with general optimization-based planning principles (e.g., as in the original Pickering et al. study⁶ from where the candidate system designs were sourced). The equity-cost indicator takes a more agnostic approach and considers that investment in a country’s investment can be a positive thing; this is the perspective in the policy for Cohesion Funding⁵.

We have clarified this nuance by adding an explanation in Methods – Equity Indicators on page 25, lines 30-33:

“Adopting the Cohesion Mechanism as the guiding cost equity principle implies that investment into national infrastructure is a positive outcome for the ensuing effects in providing low-carbon infrastructure and access to innovation; this perspective notably differs from that in the utilitarian perspective on cost (Methods – Utilitarian indicators).”

R.18 (Remarks on code availability). The code seems to run fine, just as before.
Thank you for having taken the time to test the code!

Reviewer #2 (Remarks to the Author):

R2.0 The authors have constructively engaged with and responded to my review comments. I am happy to support publication of the manuscript.

Thank you for having supported us over the course of this editorial process!

Reviewer #3 (Remarks to the Author):

R3.0 The authors mostly ignored my comments, so all of the problems with the original manuscript remain. Because the authors did not seriously engage with my suggestions, I really don't have anything else to say about the study.

Based on supplementary editor feedback, we have further clarified on page 3, lines 26-27 that our focus is only on distributive justice, as stated in the title of our manuscript: "We focus on distributive justice given its relevance to energy systems planning and that models are well-suited to assessing distributional impacts."

We further clarify our focus and write:

- Page 6, lines 14-18: "We note that there are alternative frameworks of distributive justice, and impact categories, respectively including water use and capability-based approaches⁷, relevant to energy systems development. However, these alternative frameworks and impacts are outside the scope of the present work (see Methods – Justice indicators). The aim of our work is to develop a representative case study and pave the way for future work."
- Page 23, lines 7-9: "The present study focuses on distributive justice, which is an important guiding principle in energy justice and one that is suited to model-based studies⁸. We do not try to resolve the outstanding issues with indicators for other types of justice⁹."

Nonetheless, our analysis also reveals the importance of procedural justice to achieving distributionally just systems in practice. We describe this connection at:

- Page 15, lines 1-9: "Figure 5a reveals two other practical considerations. First, achieving distributional justice in practice may depend on procedurally just processes. Here, onshore wind almost uniformly emerges as the most important technology in terms of absolute capacity for achieving a distributionally just energy system when considering the top 1% of most just candidate designs. However, onshore wind is an important technology across all candidate designs (Supplementary Fig. 17). As such, installing more onshore wind does not necessarily lead to more just outcomes (Supplementary Fig. 18). Instead, realising the distributional benefits associated with onshore wind requires clearly defining local benefits and public participation in the planning process¹."
- Page 20, 14-28: "However, while the motivation of this study is to understand how distributive justice can be used as a design criterion, our results help to highlight the real-world value of procedural justice...our modelling results highlight a particular practical dilemma pertaining to onshore wind development. According to the top 1% of most just candidate designs for the BP indicator and eight of nine interpretations of distributive justice, onshore wind is the most important technology in terms of total capacity installations. However, developing onshore wind is fraught with challenges stemming from a lack of social acceptance. Achieving distributive justice in real terms, therefore, also requires procedurally just and socially acceptable development."

With respect to Reviewer 3's most recent response, we find claim that we "mostly ignored all [the] comments" to be unsubstantiated: Indeed, we implemented a **completely new analysis** in response to prior feedback. The changes we made in response to the previous feedback round – comprising four pages of response to five questions – is summarized below in Table B. We note that some of the requested changes are not possible in the context of our study, e.g., see point 3.1.a and 3.2.c.

Thank you for considering the substantial changes made and the outlined reasoning as to why further changes are not possible in the scope of this study.

Table B. Summary of Response to Reviewers (previous editing round)

Comment	Reviewer critique	Response
3.1	a. “This omission of two of three key principles of energy justice is not just a data limitation, but a fundamental problem that undermines the entire exercise analysis”	- The focus of this study is on distributive justice, as stated in the title.- We state the focus of our work on page 3, lines 24-31- The literature lacks appropriate indicators to comment on distributive and procedural justice. It is not the goal of the present work to close these gaps.⁹ We newly state this on page 23, lines 7-9.- We write, “However, the results also highlight that a technical framework cannot resolve all justice concerns associated with a low-carbon transition... , a given infrastructure design can usually not be said to be just or unjust on its own; instead, developing just systems depends entirely on the processes in which specific pieces of infrastructure are sourced, commissioned, operated, and decommissioned.”
3.2	a. “The authors do not provide sufficient justification for the choice of ten indicators that they do use”	- We have completely revised our indicators to be more strongly based on empirically grounded theories of justice and performed an entirely new analysis based using these indicators.- The conceptual background for these indicators is shown in a brand-new Figure 1.- We explain these indicators and their development in Methods on pages 22-29.
	b. “it is quite odd that environmental protection is reduced to a single impact”	- The aim of our work is to demonstrate how such a methodology could be applied to integrate distributional justice considerations into energy systems modelling. It is not our aim to conduct a full Environmental Impact Assessment for each design.- There are data issues impeding a full environmental assessment, as argued by the Life Cycle Assessment community and presented in-text on page 21, lines 10-14 and page 23, lines 27-29.
	c. “social and cultural impacts such as loss of place attachment” are not represented by the indicators	- Understanding the social and cultural impacts, including loss of place attachment, is outside the scope of our study.- We are unaware of any ways to integrate the potential social and cultural impacts into a national-level modelling study. We invited the reviewer to provide a constructive idea on how to close this gap.
3.3	a. The equality indicators “does not redress existing disparities, which means it is not ‘just’”.	- This statement is factually incorrect. For example, egalitarian theories of justice argue that to be just is to be equal (see the Stanford Encyclopedia of Philosophy⁷).

Comment	Reviewer critique	Response
	b. “Second, relying on majority public opinion to create weights for indicators does the same thing [fails to redress existing disparities]”	- Not all theories of justice redress existing disparities. Please refer to our response to 3a (row above). - Our “balanced priorities” approach is intended to represent the public opinion and provide, “a democratically informed view of what a just energy system should resemble.” (page 16, lines 13-15)
	c. “In fact, the entire survey approach to weight the indicators seems fraught, given challenges with survey sampling.”	- We rely on survey responses to characterise the public opinion across an entire continent. “Alternative forms of public engagement, e.g., a series of public workshops, could also help elicit a more precise and deeper understanding of public opinion on the development of networked energy systems. However, workshops would be difficult to conduct on a scale as large as the EU Barometer surveys^{10,11}.” (page 28, line 32- page 29, line 3) - We rely on European Union Eurobarometer surveys, the programme for which has been running for over 50 years. Each survey we consider has over 25 000 respondents (Table 3). - We conduct an additional sensitivity analysis to test the sensitivity of our results to changing public attitudes (Supplementary Fig. 19).
	d. “the national samples completely overlook the siting challenges that come with any energy installation, where what matters the most are local public attitudes.”	- The scope of our work is on a national scale, and we use national-level energy systems modelling results. Understanding the siting challenges in greater detail is a study with a different objective, requiring another model and local opinion surveys for every municipality in Europe to investigate local attitudes towards individual siting decisions.
3.4	a. “I find the analysis itself to be poorly explained”	- Multiple readers, including the other reviewers and multiple native English speakers with non-technical background, reviewed our text and found it to be well-explained. - We addressed all additional clarification comments from the other reviewers. - For additional certainty, we checked the entire document with Grammarly.
3.5	a. “the actual implementation is problematic”	- The actual implementation has been set out as desirable by numerous scholars calling for indicator-based approaches to supporting the practical implementation of energy justice. These scholars include Prof. Benjamin Sovacool ^{9,12} , Prof. Erin Baker ⁹ , Prof. Raphael Heffron ^{12,13} , Prof. David Konisky ^{9,14} , and scholars from the United States National Laboratories ^{14,15} .

Comment	Reviewer critique	Response
		 - Our analysis relies upon candidate designs from a well-established energy system model and high-quality peer-reviewed study. Here, we build on that well-established basis.
	b. “I think [the study] fails to deliver on the desired objective of creating a methodology that can be practically applied by policymakers”	 - Model-informed policymaking has been key in energy and climate planning since the 1970s¹⁶ - We explain the steps needed to apply our methodology to a real-world process on page 22, lines 17-29.

References

1. McKenna, R. et al. System impacts of wind energy developments: Key research challenges and opportunities. *Joule* 9, 101799 (2025).
2. Mann, H. B. & Whitney, D. R. On a Test of Whether one of Two Random Variables is Stochastically Larger than the Other. *Ann. Math. Stat.* 18, 50–60 (1947).
3. European Commission. Communication from the Commission to the European Parliament, the Council, the European Economic and Social Committee, and the Committee of the Regions: EU Biodiversity Strategy for 2030. 1–23 (2020).
4. European Commission. Enhancing agricultural biodiversity. Agriculture and rural development https://agriculture.ec.europa.eu/cap-my-country/sustainability/environmental-sustainability/biodiversity_en#cap-actions (2025).
5. European Union. Regulation (EU) 2021/1058 of the European Parliament and of the Council of 24 June 2021 on the European Regional Development Fund and on the Cohesion Fund. (2021).
6. Pickering, B., Lombardi, F. & Pfenninger, S. Diversity of options to eliminate fossil fuels and reach carbon neutrality across the entire European energy system. *Joule* 6, 1253–1276 (2022).
7. Lamont, J. & Favor, C. Distributive Justice. in *The Stanford Encyclopedia of Philosophy* (ed. Zalta, E. N.) (Metaphysics Research Lab, Stanford University, 2017).
8. Lonergan, K. E., Suter, N. & Sansavini, G. Energy systems modelling for just transitions. *Energy Policy* 183, 113791 (2023).
9. Baker, E. et al. Metrics for Decision-Making in Energy Justice. *Annu. Rev. Environ. Resour.* 48, 737–760 (2023).
10. McGookin, C. et al. Advancing participatory energy systems modelling. *Energy Strategy Rev.* 52, 101319 (2024).
11. McKenna, R., Bertsch, V., Mainzer, K. & Fichtner, W. Combining local preferences with multi-criteria decision analysis and linear optimization to develop feasible energy concepts in small communities. *Eur. J. Oper. Res.* 268, 1092–1110 (2018).
12. Sovacool, B. K., Heffron, R. J., McCauley, D. & Goldthau, A. Energy decisions reframed as justice and ethical concerns. *Nat. Energy* 1, 16024 (2016).
13. Heffron, R. J., McCauley, D. & De Rubens, G. Z. Balancing the energy trilemma through the Energy Justice Metric. *Appl. Energy* 229, 1191–1201 (2018).
14. Crespo Montañés, C. et al. Enabling and centering equity and justice in clean energy transition research. *Joule* 7, 437–441 (2023).
15. Dutta, N. S. et al. JUST-R metrics for considering energy justice in early-stage energy research. *Joule* S2542435123000387 (2023) doi:10.1016/j.joule.2023.01.007.
16. Pfenninger, S., Hawkes, A. & Keirstead, J. Energy systems modeling for twenty-first century energy challenges. *Renew. Sustain. Energy Rev.* 33, 74–86 (2014).

Manuscript: NCOMMS-24-21451C

Title: Considering distributive justice as a planning principle helps navigate a diversity of future energy infrastructure designs

Response to reviewers:

Thank you very much for taking the time to review our revised manuscript and for your feedback. We appreciate your constructive contributions to improving our manuscript!

Kind regards,

The Authors

REVIEWER COMMENTS

Reviewer #1 (Remarks to the Author):

R1.0 The authors have addressed the comments, questions and suggestions they have received thoroughly and the few clarifications that I had have been resolved.

As such, I am happy with the current state of the manuscript and believe that it is suitable for publication in Nature Communications.

R1.1 Remarks on code availability:

I have reviewed the code in the initial review and after the first revision.

Thank you again for your time and support throughout the review process!